# Cryptococcal Hsf3 controls intramitochondrial ROS homeostasis by regulating the respiratory process

Xindi Gao[1], Yi Fu[1], Shengyi Sun[1], Tingyi Gu[1], Yanjian Li[1], Tianshu Sun[2,3], Hailong Li ®[4], Wei Du[1], Chenhao Suo[1], Chao Li[1], Yiru Gao[1], Yang Meng[1], Yue Ni[1], Sheng Yang[1], Tian Lan[1], Sixiang Sai[5], Jiayi Li ®[6], Kun Yu[7], Ping Wang ®[8] & Chen Ding ®[1] ✉

Mitochondrial quality control prevents accumulation of intramitochondrial-derived reactive oxygen species (mtROS), thereby protecting cells against DNA damage, genome instability, and programmed cell death. However, underlying mechanisms are incompletely understood, particularly in fungal species. Here, we show that *Cryptococcus neoformans* heat shock factor 3 (*Cn*Hsf3) exhibits an atypical function in regulating mtROS independent of the unfolded protein response. *Cn*Hsf3 acts in nuclei and mitochondria, and nuclear- and mitochondrial-targeting signals are required for its organelle-specific functions. It represses the expression of genes involved in the tricarboxylic acid cycle while promoting expression of genes involved in electron transfer chain. In addition, *Cn*Hsf3 responds to multiple intramitochondrial stresses; this response is mediated by oxidation of the cysteine residue on its DNA binding domain, which enhances DNA binding. Our results reveal a function of HSF proteins in regulating mtROS homeostasis that is independent of the unfolded protein response.

One of the critical functions of mitochondria is to drive respiration by initiating the tricarboxylic acid (TCA) cycle to trigger electron transport chain (ETC) complexes for the production of ATP through oxidative phosphorylation (OXPHOS). Beyond ATP synthesis, mitochondria are also organelles with multifaceted functionality. They play many important roles, including the generation of reactive oxygen species (ROS), the regulation of cell apoptosis, and the production of essential metabolites. Numerous studies of mitochondrial quality control (MQC) mechanisms have elucidated that the defects in mitochondrial components are causative agents for mitochondrial

dysfunction and relevant human diseases[1–4] and the critical determinants of drug resistance in human pathogens[5,6]. Upon mitochondria damage, cells consequentially activate intramitochondrial stress responses via sophisticated mechanisms that include initiating protein homeostasis and antioxidant defenses[7–9]. The mitochondrial unfolded protein response (UPR^mt), which involves gene activation of nuclear-encoded mitochondrial heat shock chaperones by nuclear heat shock factors (HSFs), is actuated to eliminate the unfolded and misfolded mitochondrial proteins[10–14]. On the other hand, the mitochondrial antioxidant defense mechanisms, consisting of superoxide

[1]College of Life and Health Sciences, Northeastern University, 110819 Shenyang, Liaoning, China. [2]Beijing Key Laboratory for Mechanisms Research and Precision Diagnosis of Invasive Fungal Diseases, 100730 Beijing, China. [3]Department of Scientific Research, Central Laboratory, Peking Union Medical College Hospital, Chinese Academy of Medical Science, 100730 Beijing, China. [4]NHC Key Laboratory of AIDS Immunology, National Clinical Research Center for Laboratory Medicine, The First Affiliated Hospital of China Medical University, 110001 Shenyang, China. [5]School of Medicine, Binzhou Medical University, 264003 Yantai, China. [6]Neural Plasticity and Repair Unit, Wallenberg Neuroscience Center, Lund University, BMC A10, 22184 Lund, Sweden. [7]College of Medicine and Biological Information Engineering, Northeastern University, 110169 Shenyang, China. [8]Department of Microbiology, Immunology, and Parasitology, Louisiana State University Health Sciences Center, New Orleans, LA 70112, USA. ✉e-mail: dingchen@mail.neu.edu.cn

dismutases, glutathione peroxidases, and peroxiredoxins, are also activated to eliminate the production of excessive ROS via electron leakage from mitochondrial complexes I and III[15,16].

HSF proteins are a family of conserved nuclear transcription factors that are master regulators of UPRs, by directly binding to the heat shock elements at the promoters of protein chaperone genes[10,17,18]. The strong therapeutic potential of HSFs enlightens the development of HSF activators or inhibitors in the treatment of neurodegenerative diseases or cancers[19–24]. While multiple HSF encoding genes were identified in the human genome, it is widely accepted that lower eukaryotic fungi such as *Saccharomyces cerevisiae* and *Candida albicans* contain a single essential gene encoding HSF[10,25–28]. In mammals, emerging evidence has suggested that Hsf1 proteins play an important role in modulating the UPR[mt] machinery under stress conditions. In the nematode *Caenorhabditis elegans*, mitochondrial stress maintains cytoplasmic proteostasis by triggering Hsf1 binding to target promoters of nuclear genes[29]. In addition, evidence indicated that the mitochondrial single-stranded DNA-binding protein 1 (SSBP1) is recruited to the nucleus, where it complexes with Hsf1 to drive the expression of chaperone genes[13]. Despite the importance of HSF proteins, their function in preserving mitochondrial integrity remains poorly understood in these model systems. Much less is understood for HSFs in the pathogenic fungal pathogen *Cryptococcus neoformans*.

Here, we report that *Cn*Hsf3, an HSF protein, modulates intramitochondrial ROS (mtROS) homeostasis in response to mitochondrial damage in *C. neoformans*. Intriguingly, the function of *Cn*Hsf3 is not directly linked to the activation of protein chaperone genes. Instead, it functions as a potent ROS-sensing regulator that detoxifies excessive mtROSs by repressing TCA genes while triggering OXPHOS gene transcription. We found that *Cn*Hsf3 contains both nuclear and mitochondrial localization signals required for its organelle-specific stress-response functions. In addition, *Cn*Hsf3 is directly activated by the oxidization of the 130-cysteine residue within its DBD ameliorates the direct binding of *Cn*Hsf3 to its target genes in mitochondria. Collectively, these findings uncovered an important regulatory mechanism of HSF proteins and shed critical insights into mechanisms of mitochondrial protection in *C. neoformans*.

## Results

### *Cn*Hsf3 regulates heat shock response via a non-UPR pathway
In contrast to *S. cerevisiae* and *C. albicans* in which the HSF protein Hsf1 is identified as a solo and essential regulator of the heat shock response (HSR)[25,26], three HSF proteins, named *Cn*Hsf1, *Cn*Hsf2 (CNAG_04176), and *Cn*Hsf3 (CNAG_04036), were identified from *C. neoformans*[30]. Quantitative RT-PCR showed that the expression of all three HSF genes is responsive to thermal stress (Fig. 1A)[30]. In addition, *Cn*Hsf3 protein expression is induced under high temperatures (Supplementary Fig. 1A). The loss of *CnHSF3*, but not *CnHSF2*, attenuates growth at 40 °C (Fig. 1B and Supplementary Fig. 1B), suggesting *Cn*Hsf2 and *Cn*Hsf3 have distinct roles in HSR. Importantly, loss of *CnHSF3* resulted in moderate but consistent attenuation in fungal pathogenicity in mice, as evidenced by prolonged survival (*P* = 0.02 and *P* = 0.03) (Supplementary Fig. 1C) and reduced pulmonary fungal burdens in animals infected with two independent *Cnhsf3Δ* mutant strains (Supplementary Fig. 1D and E). This result is consistent with a previous study in which the *Cnhsf3Δ* strain showed a reduced signature-tagged mutagenesis (STM) score in the lung tissues of A/Jcr mice[31]. These results suggested that *Cn*Hsf3 likely plays a role in protecting fungal cells from thermal stress and may also function in modulating pulmonary host immunity.

We next compared regulatory functions between *Cn*Hsf3 and *Cn*Hsf1, a conserved transcriptional regulator in response to heat shock. A galactose-induced promoter *GALP* was used to replace the endogenous *CnHSF1* promoter (Supplementary Fig. 1F and G), and the result showed that the transcription factor function of *Cn*Hsf1 is

essential (Supplementary Fig. 1H)[30], resembling that of other fungal Hsf1 proteins. This finding validates the distinction between *Cn*Hsf3 and *Cn*Hsf1.

We then further compared *Cn*Hsf3 with the known HSFs from other organisms. In *S. cerevisiae*, Skn7 possesses a Hsf1-like DBD feature that activates protein chaperone gene expression under oxidative stress. Reciprocal BLAST analysis of Skn7 in *S. cerevisiae* and *C. neoformans* genomes identified *C. neoformans* CNAG_03409 as an Skn7 homolog, different from *Cn*Hsf3. We then examined Hsf1 homologs from 31 of 37 existing fungal genomes, as well as several animal models (Fig. 1C and Supplementary Fig. 2A). Phylogenetic analysis revealed that *Cn*Hsf3 shares the highest amino acid sequence homology and comparison coverage with the human Hsf5 (hHsf5, Supplementary Fig. 2B and C). Interestingly, the expression of hHsf5 in the *Cnhsf3Δ* mutant rescued the cell growth under high temperatures (Supplementary Fig. 2D). Further analysis revealed that *Cn*Hsf3 shares sequence homology and superimposed aligned structures in the DNA binding domain (DBD) with those of hHsf1[32], with valine, instead of serine, at the DNA-interacting helix (Fig. 1D and Supplementary Fig. 2E). *Cn*Hsf3 showed fewer similarities to hHsf1 (32.41%) and hHsf2 (26.42%), compared with *Cn*Hsf1 to hHsf1 (51.85%) and hHsf2 (50.00%) (Supplementary Fig. 2F). Based on these findings, we hypothesize that *Cn*Hsf3 may exhibit unique regulatory functions and proceed to further examine these functions.

The chromatin immunoprecipitation (ChIP) assay with sequencing (ChIP-seq) analysis of DNA binding motifs using *Cn*Hsf1- and *Cn*Hsf3-FLAG proteins showed that *Cn*Hsf1 binds to the consensus nGAAn shared by HSFs of humans and *S. cerevisiae* (Supplementary Data 1). However, *Cn*Hsf3 has an adenine-rich motif (E value of $8.2 \times 10^{-44}$) (Fig. 1E), different from *Cn*Hsf1. In addition, gene ontology analysis indicated that processes regulated by *Cn*Hsf3 are remarkably disparate from those by other Hsf1 proteins. Intriguingly, without the classical regulatory features of other Hsf1, *Cn*Hsf3 is not enriched at the promoters of protein chaperones, suggesting it does not regulate the expression of these genes. Instead, *Cn*Hsf3 binds directly to promoters of genes involved in metabolism (Fig. 1F–J, Supplementary Fig. 3A and B)[30]. Remarkably, overexpressing *CnHSF3* did not induce the expression of *CnSSA1* and *CnHSP90*, in contrast to *CnHSF1* that did (Supplementary Fig. 3C). The Electrophoretic Mobility Shift Assay (EMSA) analysis demonstrated that *Cn*Hsf3 could not bind to promoters of *CnHSP70* and *CnHSP90* (Supplementary Fig. 3D). These data suggested that *Cn*Hsf3 regulates HSRs via processes not related to UPRs.

### *Cn*Hsf3 is a critical deterrent of mitochondrial metabolism and mtROS generation
We further employed RNA-Seq to decipher the possible role of *Cn*Hsf3 in the HSR of *C. neoformans*. Consistent with ChIP-seq findings, the expression of genes encoding protein chaperone showed no significant alterations (Supplementary Data 2). KEGG pathway enrichment analysis revealed significant regulatory dissimilarities in TCA and OXPHOS between the wild-type and *Cnhsf3Δ* strains (Fig. 2A, B and Supplementary Data 3). Genes of the TCA cycle that were repressed in the heat-treated wild-type cells were relatively induced in the *Cnhsf3Δ* strain (Fig. 2C). Additionally, the genes involved in OXPHOS, which remained constant in the heat-treated wild-type cells, were downregulated in the *Cnhsf3Δ* strain (Supplementary Data 2). These data implied that *Cn*Hsf3 has a role in heat response by protecting cells through modulating mitochondrial metabolic processes.

To substantiate the results from the above transcriptomic analysis, we performed a metabolomic analysis (Supplementary Data 4). The results showed a remarkable disparity in their metabolite profiles between the wild-type and *Cnhsf3Δ* strains (Fig. 2D and Supplementary Fig. 4A), consistent with transcriptomic analysis. The five key TCA cycle metabolic intermediates (citrate, malate, fumarate, isocitric acid, and α-ketoglutaric acid) were induced in the *Cnhsf3Δ* strain but no the wild-

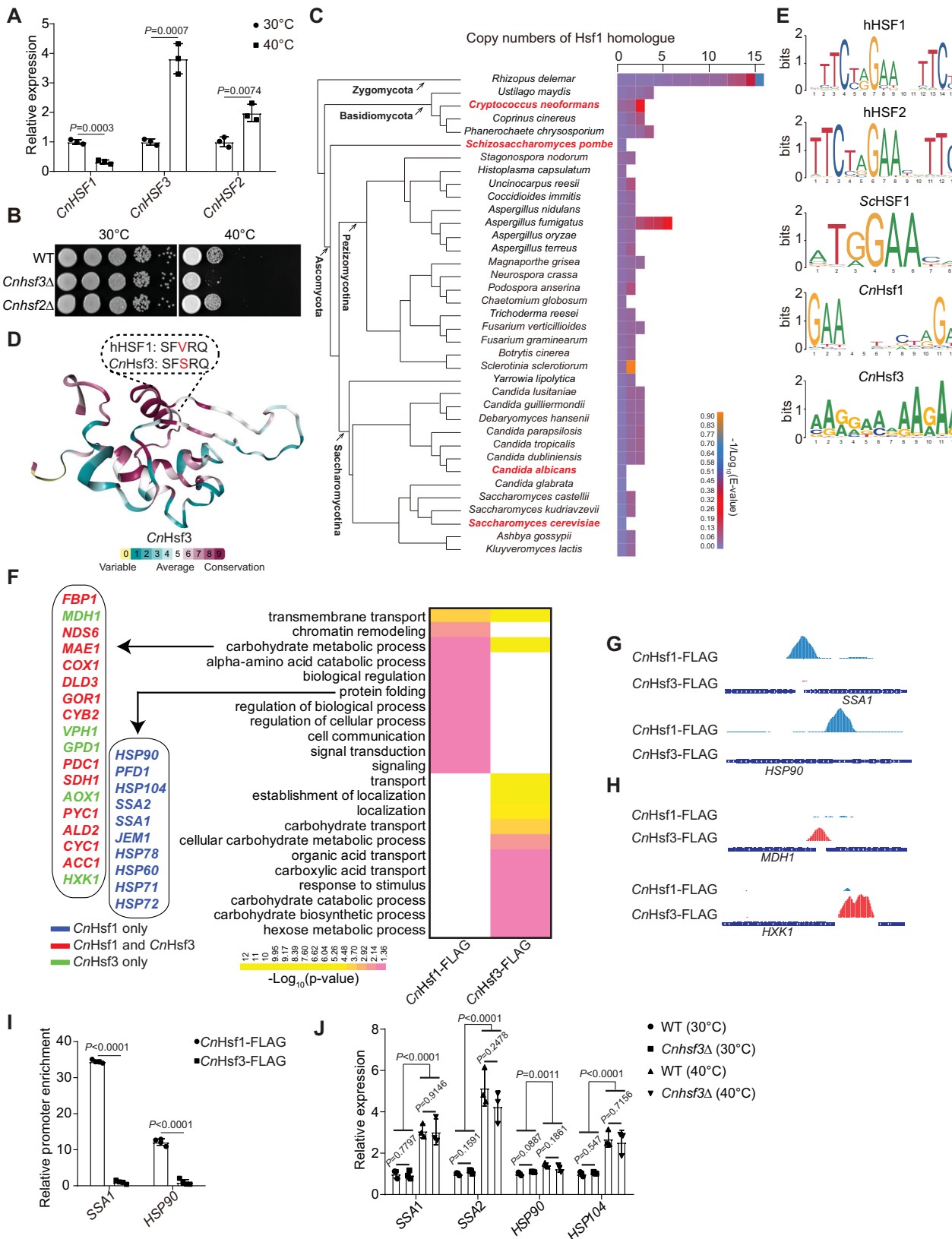

type strain (Fig. 2E, F, and Supplementary Fig. 4B). To explore whether the temperature-related growth phenotype in *Cnhsf3Δ* is induced by the TCA cycle, we bypassed the cycle by either eliminating (YNB) or substituting carbon sources (YNB supplemented with sugars) in growth media. Cells grown in the absence of six-carbon sugars were partially rescued at 40 °C (Fig. 2G and Supplementary Fig. 4C).

However, when glucose was substituted with other carbon sources, such as arabinose (a five-carbon sugar) or inositol, the temperature effect on cell growth was abrogated (Fig. 2G and Supplementary Fig. 4C).

The TCA cycle generates electron donors for ETC complexes in the mitochondria. The transcriptome data showed that 12 ETC genes

**Fig. 1 | *Cn*Hsf3 regulates the heat shock response via a mechanism not related to UPRs. A** Gene expressions of *Cryptococcus neoformans HSFs*. Wild-type cells (*n* = 3) were incubated at 30 or 40 °C for 3 h, then relative expressions were determined using qRT-PCR. *ACT1* was used as the control. Two-tailed unpaired *t*-tests were used. **B** Spotting assays of *CnHSF* mutants. Strains were spotted onto YPD agar and the plates were incubated at 30 or 40 °C for 2 days. **C** Analysis of HSF homologs in fungi. The model fungi *Cryptococcus neoformans, Schizosaccharomyces pombe, Saccharomyces cerevisiae*, and *Candida albicans* are shown in red. **D** Homologous modeling of the DBDs of *Cn*Hsf3 and hHsf1. The *Cn*Hsf3 DBD structure was predicted and compared with that of hHsf1 using ConSurf software[32]. **E** Comparative analysis of HSF binding motifs. Binding motifs were calculated using the MEME Suite. *Cn C. neoformans, Sc S. cerevisiae*, h human. **F** Gene ontology analyses of ChIP-seq results of 40 °C-treated cells. Genes regulated by *C.* *neoformans* Hsf1 or Hsf3 are shown in blue or green, respectively. Genes co-regulated by *Cn*Hsf1 and *Cn*Hsf3 are shown in red. **G** Illustration of *Cn*Hsf binding to protein chaperone gene promoters. ChIP-seq data of the protein chaperones *CnSSA1* and *CnHSP90* are shown. **H** Illustration of *Cn*Hsf binding to carbohydrate metabolism gene promoters. ChIP-seq data of *MDH1* and *HXK1* are shown. **I** ChIP-PCR analyses of protein chaperone gene promoters. *CnHSF1*-FLAG (*n* = 4) and *CnHSF3*-FLAG (*n* = 4) strains were incubated at 40 °C for 3 h, then ChIP-PCR was used to measure the enrichment of the promoter sequences. Data of *CnHSF3*-FLAG was used as the reference. A two-tailed unpaired *t*-test was used. **J** qRT-PCR results of protein chaperone genes. The wildtype (*n* = 3) and *Cnhsf3Δ* (*n* = 3) strains were incubated at 30 or 40 °C for 3 h. WT (30 °C) was used as the reference. Two-tailed unpaired *t*-tests and two-way ANONA were used. Data are expressed as mean ± SD. Source data are provided as a Source Data file.

were repressed in *Cnhsf3Δ*: five of eight complex I genes (Fig. 3A and Supplementary Fig. 5A), *CNAG_05633* and *CNAG_02938* from complexes III and IV, and 7 of 10 complex V genes (Fig. 3B and Supplementary Fig. 5B). Because ETC complex genes are encoded in both the nuclear and mitochondrial genomes and the transcriptome analysis utilized oligo(dT) for reverse transcription that could result in the insufficient synthesis of mitochondrial cDNA (Supplementary Fig. 5C)[33], we employed random hexamer qRT-PCRs to measure mitochondrial encoding ETC gene expression. Results showed that gene expression of six of seven tested ETC genes is regulated by *Cn*Hsf3 at 40 °C (Fig. 3C and Supplementary Fig. 5A and B). Interestingly, *Cn*Hsf3-FLAG ChIP-PCR assays showed a direct binding to the regulatory regions of these mitochondrial encoding genes at 40 °C (Fig. 3D), indicating that *Cn*Hsf3 is both a nuclear and mitochondrial targeting transcription factor.

To verify that the heat shock phenotype in *Cnhsf3Δ* is a result of diminished expression of ETC genes, we complemented the *Cnhsf3Δ* strain with integrative plasmids harboring *Cn*Hsf3-regulated ETC genes, including *QCR9* (ubiquinol cytochrome-c reductase 9, complex III), *NDUFA5* (NADH:ubiquinone oxidoreductase subunit A5, complex I), *NDUFS6* (NADH:ubiquinone oxidoreductase subunit S6, complex I), and *CNAG_09000* (NADH-ubiquinone oxidoreductase chain 1, mitochondrial encoded) (Supplementary Fig. 5D–F). Cells expressing *QCR9* and *NDUFA6* failed to restore cell growth (Supplementary Fig. 5G–J). Tolerance to heat shock was found when *QCR9* and *NDUFA5* or *QCR9* and *CNAG_09000* were co-expressed (Fig. 3E and Supplementary Fig. 5J). These results demonstrated that *Cn*Hsf3 defends temperature tolerance by regulating the subunit genes of both complex I and complex III. Impairment in ETC resulted in reduced activities of NADH dehydrogenase and ATP synthesis (Fig. 3F and G).

The defects in complexes I and III prompted us to test whether *CnHSF3* deletion would cause excessive mtROS production. Employing a mitochondria-specific ROS indicator, MitoSOX, we showed that, at 40 °C, *Cnhsf3Δ* has about 10% more mtROS-producing cells than the wild-type strain (Fig. 3H). The induction and the increase in signal intensity occurred at an early stage of heat shock (within 5 min) (Fig. 3I and J). Furthermore, an increase in ROS-producing cells and cytosolic ROS was also detected (Fig. 3K and Supplementary Fig. 6A). These results suggested that the lack of *Cn*Hsf3 regulation attenuates ETC activities, leading to mitochondrial dysfunction and mtROS overload and that complexes I and III play important roles in this process. This mtROS overload then results in mitochondrial genome instability (Supplementary Fig. 6B and C).

To further examine the effects of mtROS overload, we overexpressed mitochondrial superoxide dismutase genes *SOD1* (localized to the cytosol and mitochondrial intermembrane space) and *SOD2* (mitochondrial matrix), respectively[34,35] (Supplementary Fig. 6D). The result showed that *Cnhsf3Δ* cells harboring mitochondrial *SOD2*, but not *SOD1*, integrative plasmid restored cell growth (Fig. 3L and Supplementary Fig. 6E). Furthermore, supplementing *Cnhsf3Δ* cells with the ROS scavenger N-acetyl-L-cysteine (NAC) provided protection

similarly to *SOD2* overexpression (Fig. 3M and Supplementary Fig. 6F). In consistence with the above results, overexpressing *QCR9* and *NDUFA5* significantly reduced mtROS production (Supplementary Fig. 6G and H). These data extensively demonstrated that *Cn*Hsf3 plays an important role in mitochondrial protection by modulating its metabolic processes. *Cn*Hsf3 responds to ROS overload of the mitochondrial matrix rather than an elevated temperature.

### *Cn*Hsf3 is a mitochondrion targeting transcription factor
ChIP-seq and ChIP-PCR analysis indicated that *Cn*Hsf3 is a mitochondrial transcription factor since it both binds to and regulates genes encoded in the mitochondrial genome (Figs. 3D and 4A). For verification, we performed a *Cn*Hsf3-FLAG IP experiment followed by mass spectrometry analysis. The data showed that *Cn*Hsf3 interacts with a group of mitochondria-specific proteins (Fig. 4B and Supplementary Data 5). Immunoblotting using proteins isolated from mitochondria indicated the presence of *Cn*Hsf3-FLAG (Fig. 4C). The translocase of the inner membrane (Tim) proteins is localized in the mitochondrial intermembrane space and functions as essential chaperones for importing proteins. Indeed, *Cn*Hsf3-FLAG is a client protein of Tim44 via a protein–protein interaction (Fig. 4D). The relative protein intensity ratio of *Cn*Hsf3 and Tim44 remained constant as temperature increases (Supplementary Fig. 7A), implying that mitochondrial trafficking of *Cn*Hsf3 is not regulated by temperature. A mitochondrial targeting signal (MTS) was predicted in *Cn*Hsf3 (residues 1–10) whose deletion abolished the mitochondrial localization and protein interaction with Tim44 (Fig. 4E and F), in addition to impaired growth (Fig. 4G and Supplementary Fig. 7B). To further test the mitochondrial localization of *Cn*Hsf3, strains harboring Tim44-mCherry and *Cn*Hsf3-GFP or *Cn*Hsf3 (MTS^mut)-GFP plasmids were constructed. Fluorescent signals of *Cn*Hsf3-GFP demonstrated both nuclear and cytosolic localizations. Fluorescence of *Cn*Hsf3-GFP was partially merged with that of Tim44-mCherry, and *Cn*Hsf3 (MTS^mut)-GFP showed predominantly nucleus localization (Supplementary Fig. 7C).

Our data have demonstrated the crucial functions of *Cn*Hsf3 in maintaining mitochondrial metabolic processes and serving as a key regulator of mtROS homeostasis. In addition, disrupting *CnHSF3* function recasts mitochondrial morphology, with more individual structures and an equal amount of networks (Fig. 4H and I), resulting in more individually fragmented and less tubular mitochondria (Fig. 4J and K). Moreover, a loss in mitochondrial membrane potential was detected in *Cnhsf3Δ* cells using tetramethylrhodamine methyl ester staining (Fig. 4L).

### Nuclear localization signal (NLS) and mitochondrial targeting sequence (MTS) are required for *Cn*Hsf3 function
We have demonstrated that *Cn*Hsf3 is an important modulator in mtROS by regulating the expression of the TCA cycle and ETC genes. The latter are both nuclear and mitochondrial encoded. Given that *Cn*Hsf3 is functional in both organelles, we investigated its binary functionality. Charged residues found on the putative NLS (residues

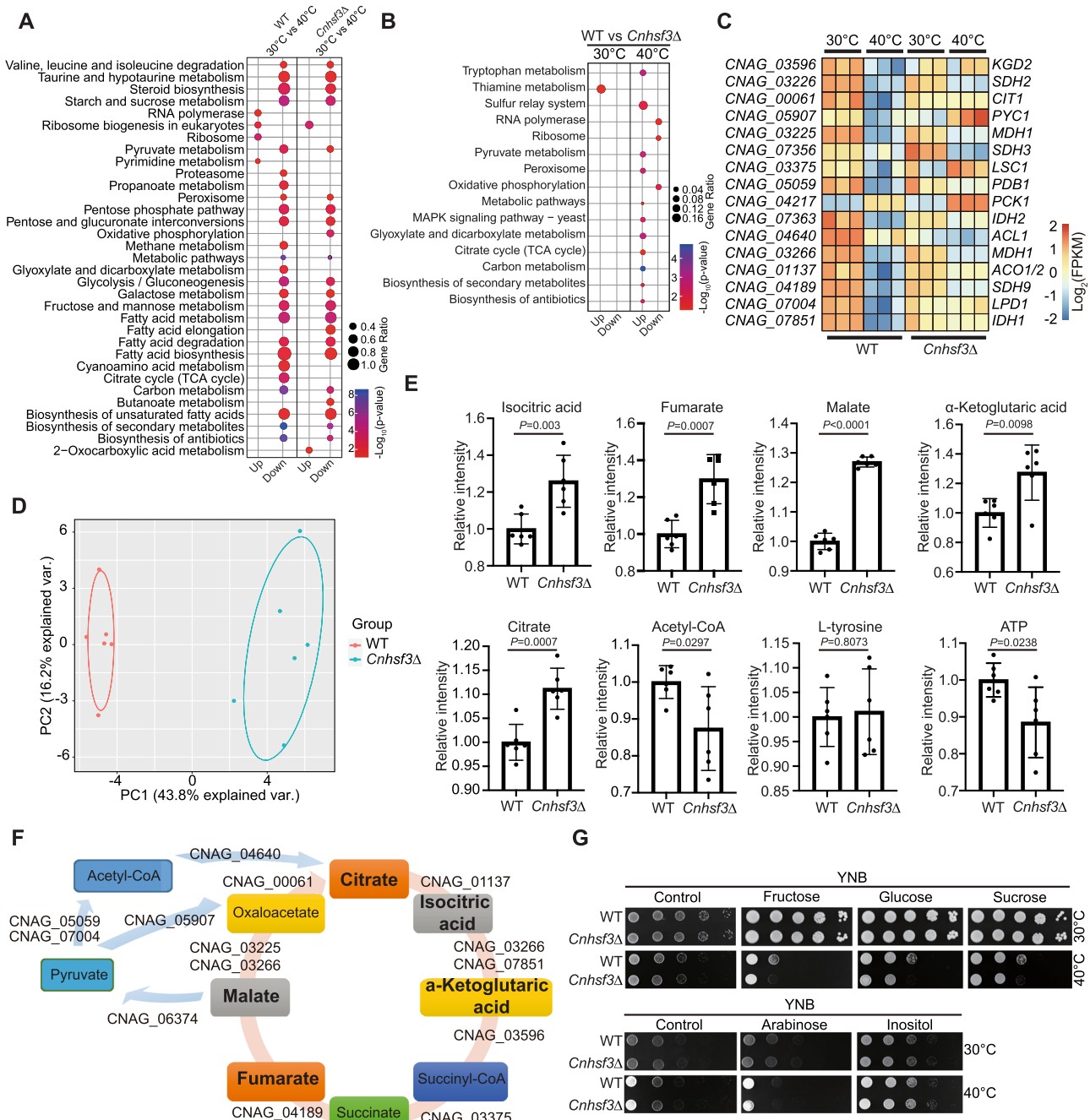

**Fig. 2 | CnHsf3 regulates the heat shock response via a metabolic pathway.**
**A** Kyoto Encyclopedia of Genes and Genomes (KEGG) analyses. KEGG analyses were performed using RNA-seq data from wildtype strains (30 versus 40 °C) and the *Cnhsf3Δ* strain (30 versus 40 °C). **B** Comparisons of KEGG analyses of wildtype and *Cnhsf3Δ* strains. Transcriptome data from the two strains at 30 and 40 °C were compared. **C** Transcriptome heatmap of TCA cycle genes. Transcriptome data of 16 genes involved in the TCA cycle were compared. **D** Principle component analysis between the metabolomes of wildtype (*n* = 6) and *Cnhsf3Δ* (*n* = 6) strains. The analysis was determined using the devtool and ggbiplot package. **E** Metabolome

data from wildtype and *Cnhsf3Δ* strains. Five TCA cycle intermediates (isocitric acid, fumarate, malate, α-ketoglutaric acid, and citrate), acetyl-CoA, ATP, and L-tyrosine are shown (*n* = 6). Two-tailed unpaired *t*-tests were used. **F** Illustrated metabolome data for the TCA cycle. Key enzymes are shown with gene IDs. Bold text indicates induction. **G** Spotting assays in various carbon media. Wildtype and *Cnhsf3Δ* cells were spotted onto YNB plates supplemented with the carbon sources indicated. The control was provided with no sugar supplement. Data are expressed as mean ± SD. Source data are provided as a Source Data file.

40−49) and MTS (residues 1−10) were mutated to generate *CnHSF3-NLS*^mut and *CnHSF3-MTS*^mut strains within the context of the FLAG-tagged proteins (Fig. 5A). The *CnHSF3-MTS*^mut strain showed no mitochondrial localization at either temperature and it did not interact with Tim44 (Fig. 5B and C). The *Cn*Hsf3-NLS^mut-GFP protein was found to be excluded from the nucleus (Fig. 5D). Additionally, *Cn*Hsf3 target genes encoded in the nuclei and mitochondria were analyzed using qRT-PCR

and the results showed that *Cn*Hsf3-MTS^mut is capable of activating *QCR9* gene expression but not mitochondrial-encoded *CNAG_09000*, whereas *Cn*Hsf3-NLS^mut demonstrated a regulation pattern reciprocal to that of *Cn*Hsf3-MTS^mut (Fig. 5E).

We further dissected *Cn*Hsf3 functionality into that of the nuclei and the mitochondria by challenging cells with high temperatures. Both mutants failed to complement *Cnhsf3Δ* growth defects at 40 °C,

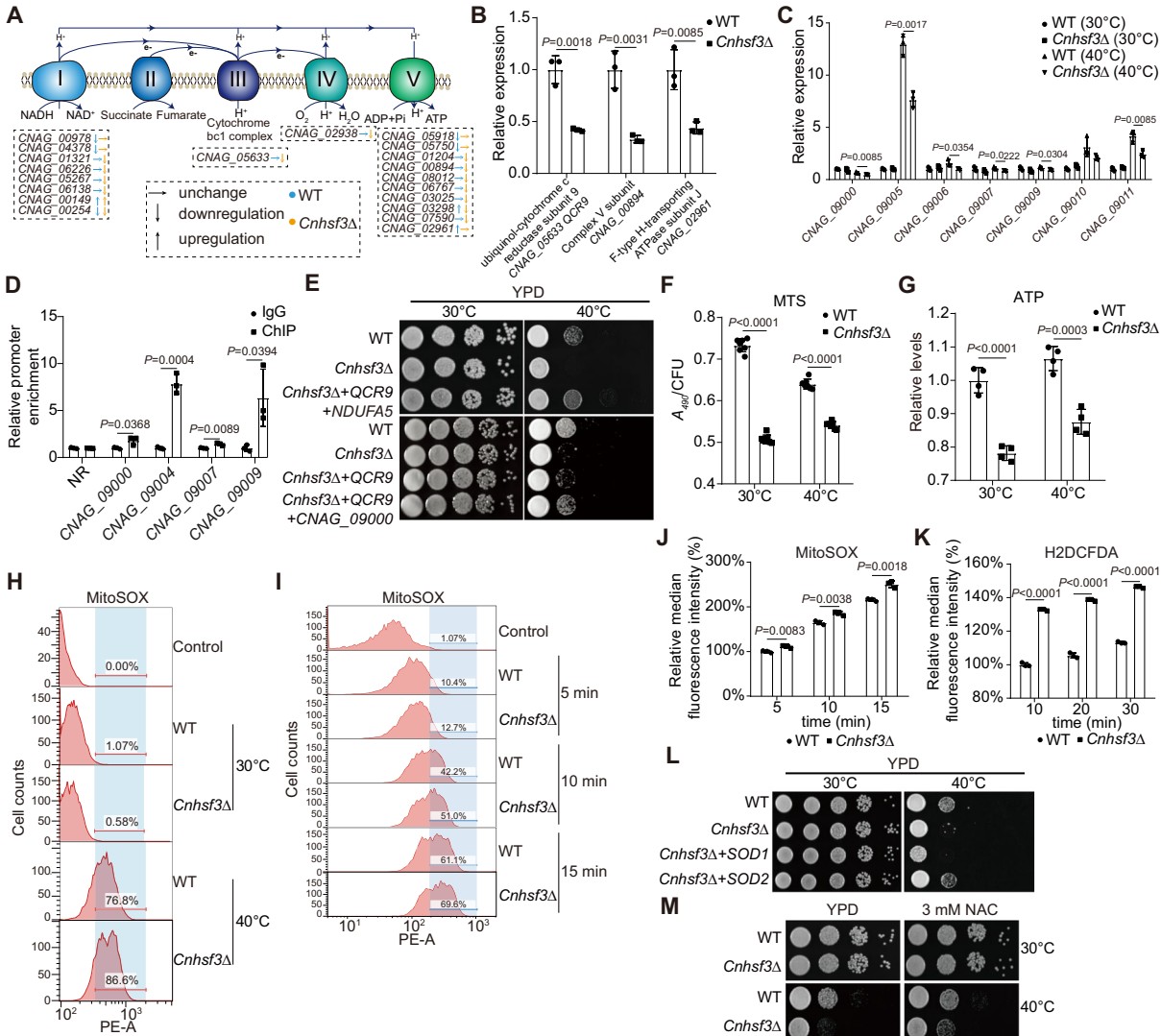

**Fig. 3 | CnHsf3 represses mtROS production by activating ETC gene expression.**
**A** ETC gene regulation scheme in *Cnhsf3Δ* from RNA-seq results. Regulation patterns of the wildtype strain are shown in blue; those of *Cnhsf3Δ* in orange. **B** qRT-PCR results of ETC genes. Analyses were performed using RNA samples from heat-shocked *Cnhsf3Δ* cells and wildtype cells (*n* = 3). **C** qRT-PCR results of mitochondrial encoding ETC genes. Analyses were performed as described in (**B**) (*n* = 3). **D** ChIP-PCR results of *Cn*Hsf3-FLAG. The *CnHSF3-FLAG* strain (*n* = 3) was incubated at 40 °C, then ChIP-PCR was performed. NR (nonregulated mitochondrial region by *Cn*Hsf3) was used as control. **E** Spotting assays of *CnHSF3* mutant strains. Five strains (wildtype, *Cnhsf3Δ*, *QCR9*-overexpressing *Cnhsf3Δ*, *QCR9*-and-*NDUFA5*-overexpressing *Cnhsf3Δ*, and *QCR9*-and-*CNAG_09000*-overexpressing *Cnhsf3Δ*) were spotted onto YPD agar, then incubated at 30 or 40 °C for 2 days. **F** Measures of complex I activity (MTS). Wild type and *Cnhsf3Δ* cells (*n* = 7) were incubated at 30 or 40 °C, then the activity of complex I was measured using MTS tetrazolium

compound. **G** ATP levels. Cells (*n* = 4) were prepared as described in **E**, then ATP levels were measured. **H** Quantification of cell populations stained with MitoSOX. Wild type and *Cnhsf3Δ* cells were loaded with MitoSOX and incubated at 30 or 40 °C for 30 min, then flow cytometry was performed. **I** Time course of mtROS production. Wild type and *Cnhsf3Δ* strains were incubated at 40 °C for 5, 10, or 15 min, then staining and flow cytometry were performed. **J** Quantification of MitoSOX signals. Fluorescence signals (*n* = 3) were quantified and plotted. **K** Quantification of cytosolic ROSs. Wild type and *Cnhsf3Δ* (*n* = 3) cells were loaded with H2DCFDA and incubated at 40 °C for 10, 20, or 30 min, then flow cytometry was performed. **L** Spotting assays of *SOD* overexpression. *SOD1*-or *SOD2*-overexpressing strains were spotted and incubated at 30 or 40 °C for 2 days. **M** Spotting assay of *Cnhsf3Δ*. Indicated strains were spotted with or without 3 mM NAC, then the plates were incubated at 30 or 40 °C for 2 days. Data are expressed as mean ± SD. Two-tailed unpaired *t*-tests were used. Source data are provided as a Source Data file.

in contrast to the wild-type *CnHSF3* gene (Fig. 5F). mtROS-producing cell counts and signal intensities of these mutants were significantly greater than those of the wild-type cells (Fig. 5G and H). These findings strongly suggested that *Cn*Hsf3 is simultaneously targeted to nuclei and mitochondria to activate ETC gene transcription that detoxifies mtROSs.

### *Cn*Hsf3 governs the universal mitochondrial stress response
Given that *Cn*Hsf3 protects mitochondria from mtROS overload, not high temperatures (Fig. 3L and M), we determined whether other mtROS inducers mimic the HSR phenotypes of *Cn*Hsf3. Treating Cn*hsf3Δ* cells with antimycin A, a complex III inhibitor, resulted in

growth impairment, even at 30 °C, which resembled the growth phenotype of *Cnhsf3Δ* at 40 °C (Fig. 6A and Supplementary Fig. 8A). Induced *CnHSF3* gene expression was time-dependent under treatment with antimycin A (Fig. 6B). Moreover, mitochondrial ETC gene expression reduction was detected for *CNAG_09000*, *CNAG_09005*, *CNAG_09006*, and *CNAG_09011* (Fig. 6C). *Cn*Hsf3 bindings to promoters of *CNAG_09000*, *CNAG_09004*, *CNAG_09007* and *CNAG_09009* were significantly enriched (Fig. 6D). Additionally, in response to antimycin A treatment, nuclear-encoded *QCR9* expression was also reduced in *Cnhsf3Δ* cells (Supplementary Fig. 8B) but both mitochondrial and cytosolic ROS generation was increased (Supplementary Fig. 8C–F). Therefore, antimycin A treatment mimics the

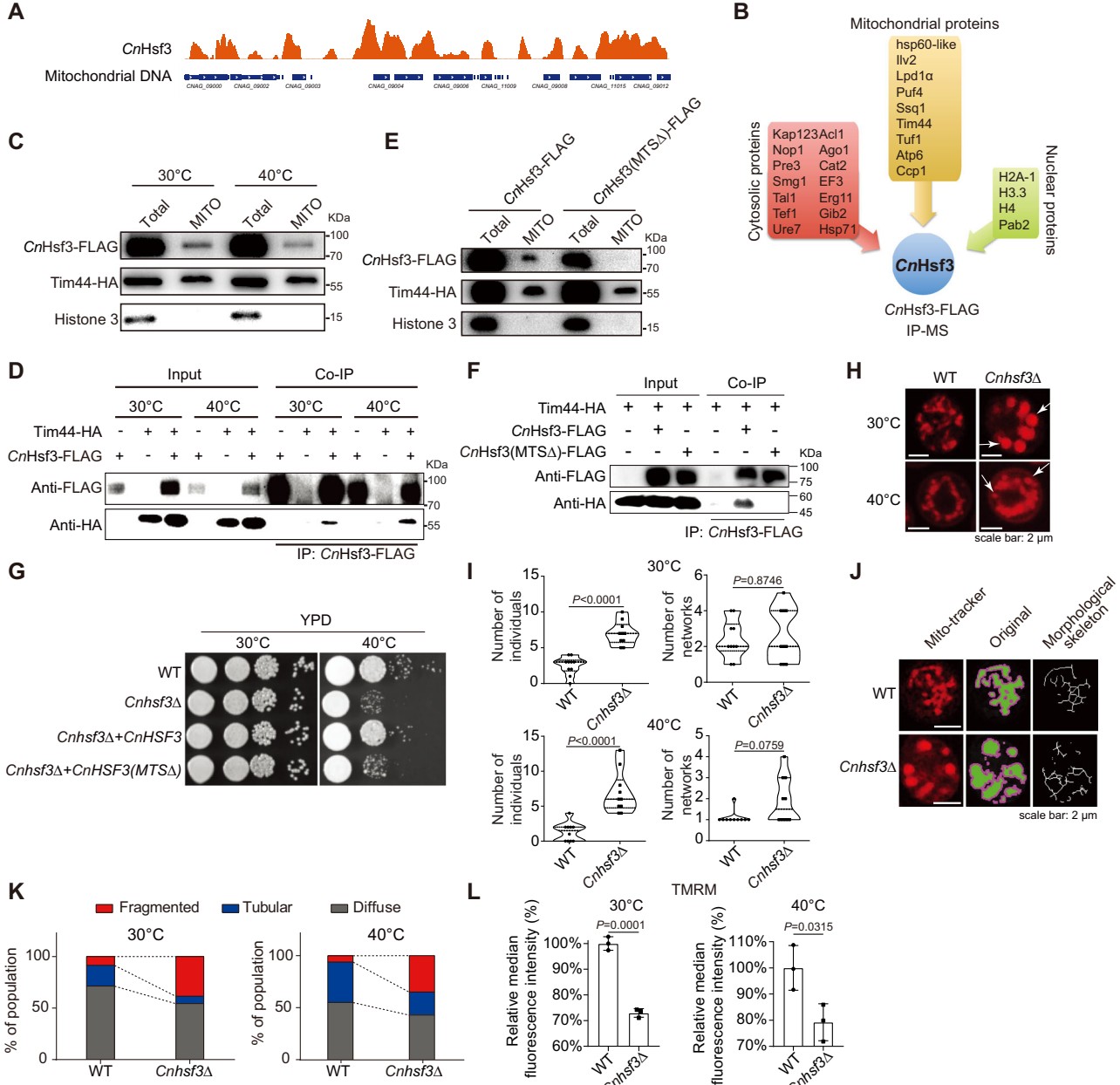

**Fig. 4 | CnHsf3 is a mitochondrial targeting transcription factor. A** Scheme of CnHsf3-FLAG binding on mitochondrial DNA. **B** CnHsf3-FLAG protein isolated by co-IP followed by mass spectrometry. CnHsf3-FLAG interacting proteins are shown. **C** Mitochondria isolation and immunoblots. Mitochondria isolated from cells grown at 30 or 40 °C were analyzed using immunoblotting. Data were obtained from three independent experiments and representative images are shown. **D** CnHsf3-FLAG and Tim44-HA co-IP immunoblots. The *TIM44-HA/CnHSF3-FLAG*, *TIM44-HA*, and *CnHSF3-FLAG* strains were grown at 30 or 40 °C, protein co-IP was performed. Data were obtained from two independent experiments and representative images are shown. **E** Immunoblots of the *CnHSF3 (MTSΔ)-FLAG* strain. Mitochondrial proteins at 40 °C were analyzed using immunoblotting. Data were obtained from three independent experiments and representative images are shown. **F** CnHsf3 (MTSΔ)-FLAG and Tim44-HA co-IP immunoblots. The *TIM44-HA/CnHSF3(MTSΔ)-FLAG*, *TIM44-HA*, and *TIM44-HA/CnHSF3-FLAG* strains were grown at 40 °C, protein co-IP was performed. Data were obtained from three independent experiments and representative images are shown. **G** Spotting assay of the

*CnHSF3(MTSΔ)-FLAG* strain. Indicated strains were spotted and incubated at 30 or 40 °C. **H** Mitochondrial morphological analyses. Wild type and *Cnhsf3Δ* cells were grown at 30 or 40 °C for 3 h, then stained with MitoTracker and observed for mitochondrial morphology. Data were obtained from three independent experiments and representative images are shown. **I** Quantifying the mitochondrial structures. The ImageJ MiNA toolset was used to count individual and network mitochondria in indicated cells ($n=10$). Two-tailed Mann–Whitney tests were used. **J** Identification of mitochondrial structures. Microscopic photographs in Fig. 3H were analyzed using the ImageJ MiNA toolset. Data were obtained from three independent experiments and representative images are shown. **K** Mitochondrial structure distribution analyses. The mitochondrial morphologies were counted from 150 cells in a blinded manner. Percentages were calculated. **L** Evaluation of mitochondrial membrane potentials. Wild type and *Cnhsf3Δ* cells ($n = 3$) were grown at 30 or 40 °C for 30 min, then tetramethylrhodamine was used, and percentages were calculated. Two-tailed unpaired *t*-tests were used. Data are expressed as mean ± SD. Source data are provided as a Source Data file.

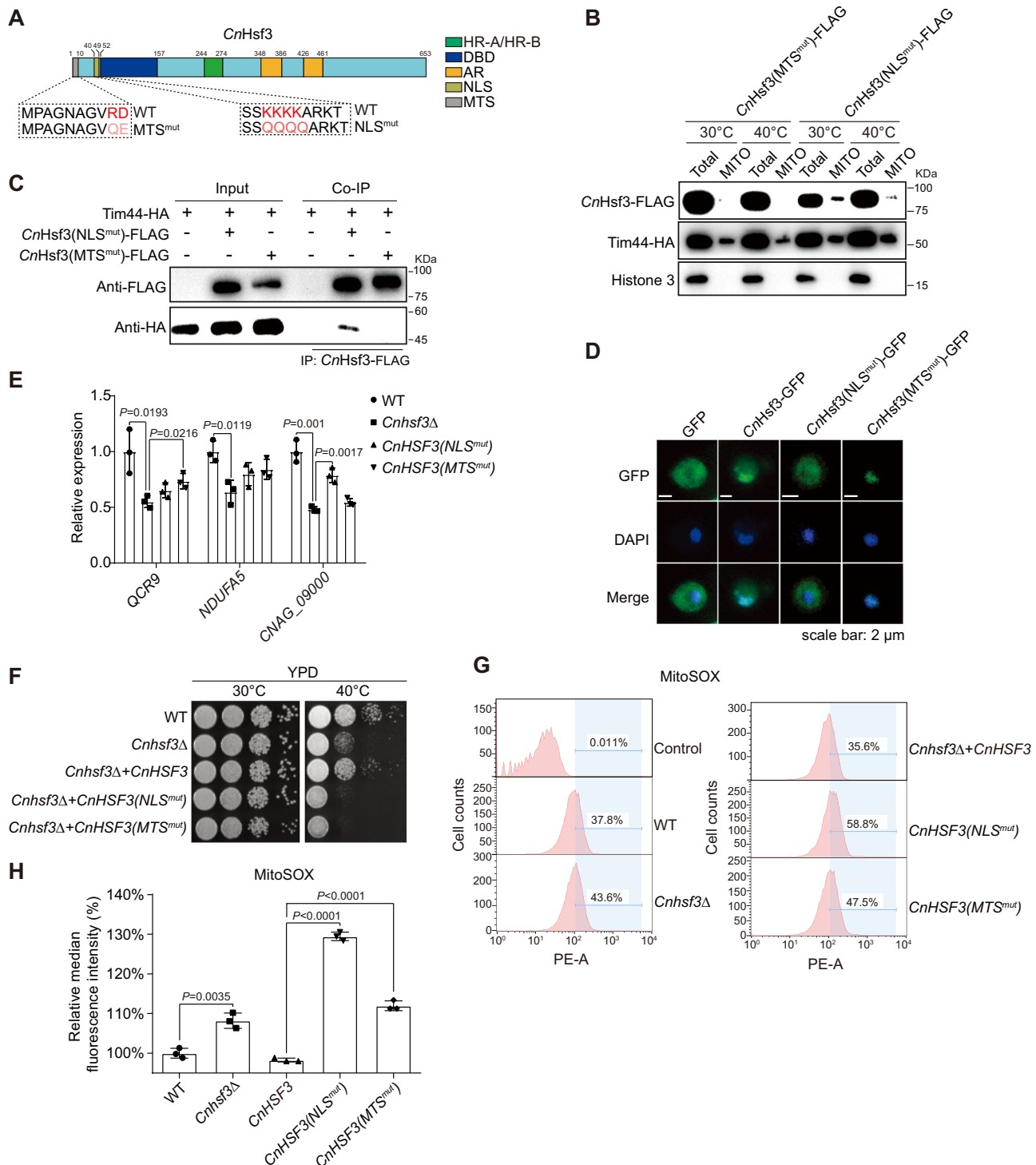

**Fig. 5 | Both nuclear and mitochondrial targeting signals are required for CnHsf3 function. A** Schemes of *Cn*Hsf3 NLS and MTS. Lysine residues (42–45) of NLS were mutated to glutamine. Arginine or aspartic acid residues (9–10) of MTS were mutated to glutamine or glutamic acid, respectively. **B** Detection of *Cn*Hsf3 (MTS^mut) in mitochondria. Mitochondria were isolated and analyzed using immunoblotting. Data were obtained from three independent experiments and representative images are shown. **C** Protein co-IP of *Cn*Hsf3 (MTS^mut)-FLAG and Tim44-HA. Indicated fungal strains were grown at 40 °C for 3 h, then co-IP was performed. Data were obtained from three independent experiments and representative images are shown. **D** Localization of *Cn*Hsf3 (NLS^mut)-GFP. Indicated strains were incubated at 40 °C for 3 h, then stained with DAPI followed by microscopic

analyses. **E** Quantification of *Cn*Hsf3 target genes in *CnHSF3* mutants. *QCR9*, *NDUFA5*, and *CNAG_09000* in RNA isolated from wildtype, *Cnhsf3Δ*, *CnHSF3* (*NLS^mut*), and *CnHSF3* (*MTS^mut*) strains at 40 °C were quantified using qRT-PCR. **F** Spotting assays of *Cn*HSF mutants. Indicated strains were grown and spotted onto YPD agar, then incubated at 30 or 40 °C for 2 days. **G** Cell population analysis of MitoSOX-stained *CnHSF3* mutants. Wild type, *Cnhsf3Δ*, *CnHSF3*, *CnHSF3* (*NLS^mut*), and *CnHSF3* (*MTS^mut*) strains were incubated at 40 °C for 30 min, then MitoSOX-based flow cytometry was performed. **H** Quantification of MitoSOX. Fluorescence signals were quantified and plotted. Data are expressed as mean ± SD (*n* = 3). Two-tailed unpaired *t*-tests were used. Source data are provided as a Source Data file.

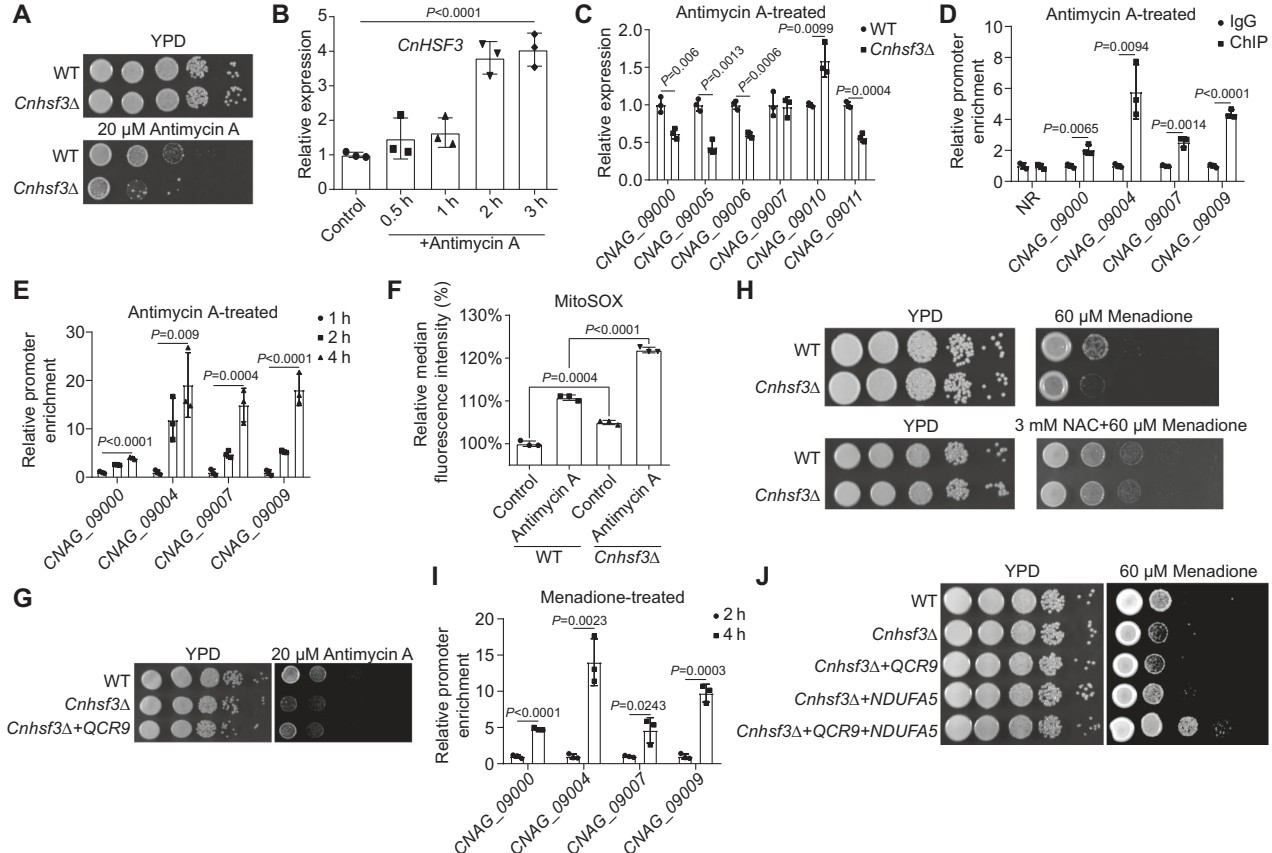

**Fig. 6 | *Cn*Hsf3 protects mitochondria from oxidative stress. A** Effect of antimycin A on cell growth in *Cnhsf3Δ*. **B** *CnHSF3* expression in response to antimycin A. Wildtype cells (*n* = 3) were treated with 10 μM antimycin A, then after 0.5, 1, 2, or 3 h of incubation, RNA was isolated, and qRT-PCR was used to quantify *CnHSF3*. One-way ANOVA tests were used. **C** Quantification of *Cn*Hsf3 target genes. RNA isolates from wildtype and *Cnhsf3Δ* strains (*n* = 3) treated with 10 μM antimycin A for 3 h were analyzed to quantify the indicated *Cn*Hsf3 target genes, determining gene expression. Two-tailed unpaired *t*-tests were used. **D** ChIP-PCR for *Cn*Hsf3 target genes in response to antimycin A. Cells (*n* = 3) were treated as described in **C**, then ChIP-PCR was performed. An NR (nonregulated mitochondrial region by *Cn*Hsf3) was used as a negative control. Two-tailed unpaired *t*-tests were used. **E** ChIP-PCR for *Cn*Hsf3 target genes. Cells (*n* = 3) were incubated with 10 μM antimycin A for 1, 2, or 4 h, then ChIP-PCR was performed. One-way ANOVA tests were used.

**F** Antimycin A induces mtROSs. Wildtype and *Cnhsf3Δ* (*n* = 3) cells were treated with 10 μM antimycin A, then stained with MitoSOX, and flow cytometry was performed to quantify the signals. Two-tailed unpaired *t*-tests were used. **G** Spotting assay of the *QCR9*-overexpressing strain. This strain was spotted onto YPD agar with or without 20 μM antimycin A, then was incubated at 30 °C for 2 days. **H** Treatment of *Cnhsf3Δ* with menadione. Wild type and *Cnhsf3Δ* cells were spotted onto YPD agar with or without 60 μM menadione, then were incubated at 30 °C for 2 days. **I** ChIP-PCR of *Cn*Hsf3 target genes. Cells (*n* = 3) were treated with 30 μM menadione for 2 or 4 h, then ChIP-PCR was performed. Two-tailed unpaired *t*-tests were used. **J** Spotting assay of the *QCR9* and *NDUFA5* overexpressing strains. Indicated strains were spotted onto YPD agar with or without menadione, then were incubated at 30 °C for 2 days. Data are expressed as mean ± SD. Source data are provided as a Source Data file.

regulation patterns of gene expression and mtROS production, and cell growth phenotypes under high temperatures. In the mitochondrial genome, *Cn*Hsf3-FLAG was significantly enriched, and the binding was enhanced when antimycin A treatment was prolonged (Fig. 6E). Antimycin A is a potent superoxide generator via inhibition of complex III (Supplementary Fig. 5B). In this way, it further provoked mtROS production in *Cnhsf3Δ* cells (Fig. 6F), and complementation with the complex III subunit (*QCR9*) fully rescued cell growth impairment in the *Cnhsf3Δ* strain (Fig. 6G and Supplementary Fig. 8A).

Moreover, treatment with another mitochondrial oxidative stress inducer, menadione, resulted in a result similar to that of antimycin A: mtROS production was elevated, and *CnHSF3* gene expression was activated (Supplementary Fig. 8G and H). Again, *Cnhsf3Δ* cell growth was impaired but could be rescued by supplementing with NAC (Fig. 6H and Supplementary Fig. 8I, J). Menadione triggers the enrichment of *Cn*Hsf3 in the mitochondrial genome (Fig. 6I), compared with untreated ChIP-PCR (Supplementary Fig. 8K), and bindings at four mitochondrial genes were ameliorated in a time-dependent manner (Fig. 6I). The regulation of *CNAG_09000* by *Cn*Hsf3 in response to antimycin A or menadione resembled the regulation

pattern by heat shock (Fig. 6E, I and Supplementary Fig. 8L). Even though complementation with *NDUFA5* restored *Cnhsf3Δ* cell growth, simultaneous overexpression of *NDUFA5* and *QCR9* resulted in a wild-type phenotype (Fig. 6J and Supplementary Fig. 8M). Taken together, these data showed that *Cn*Hsf3 is a sensor and modulator of mtROS homeostasis under various conditions that lead to mitochondrial dysfunction.

## Oxidation of cysteine residues on DBDs ameliorates *Cn*Hsf3 binding to the mitochondrial genome

We have demonstrated that the accumulation of mtROSs activates *Cn*Hsf3 function and enhances the binding of *Cn*Hsf3 to mitochondrial DNA (Fig. 6E and I). We have also shown that overexpression of the mtROS detoxifier Sod2 markedly reduces the binding of *Cn*Hsf3 to mitochondrially encoded target genes (Fig. 7A). We then hypothesize that *Cn*Hsf3 could be activated by direct oxidization on the DNA binding domain (DBD). To test this hypothesis, we analyzed the *Cn*Hsf3 protein electrophoretic mobility in the absence of a reducing agent after incubation at 30 or 40 °C. A finding of higher-molecular-weight *Cn*Hsf3 (Fig. 7B) implied that *Cn*Hsf3 could be oxidized at high

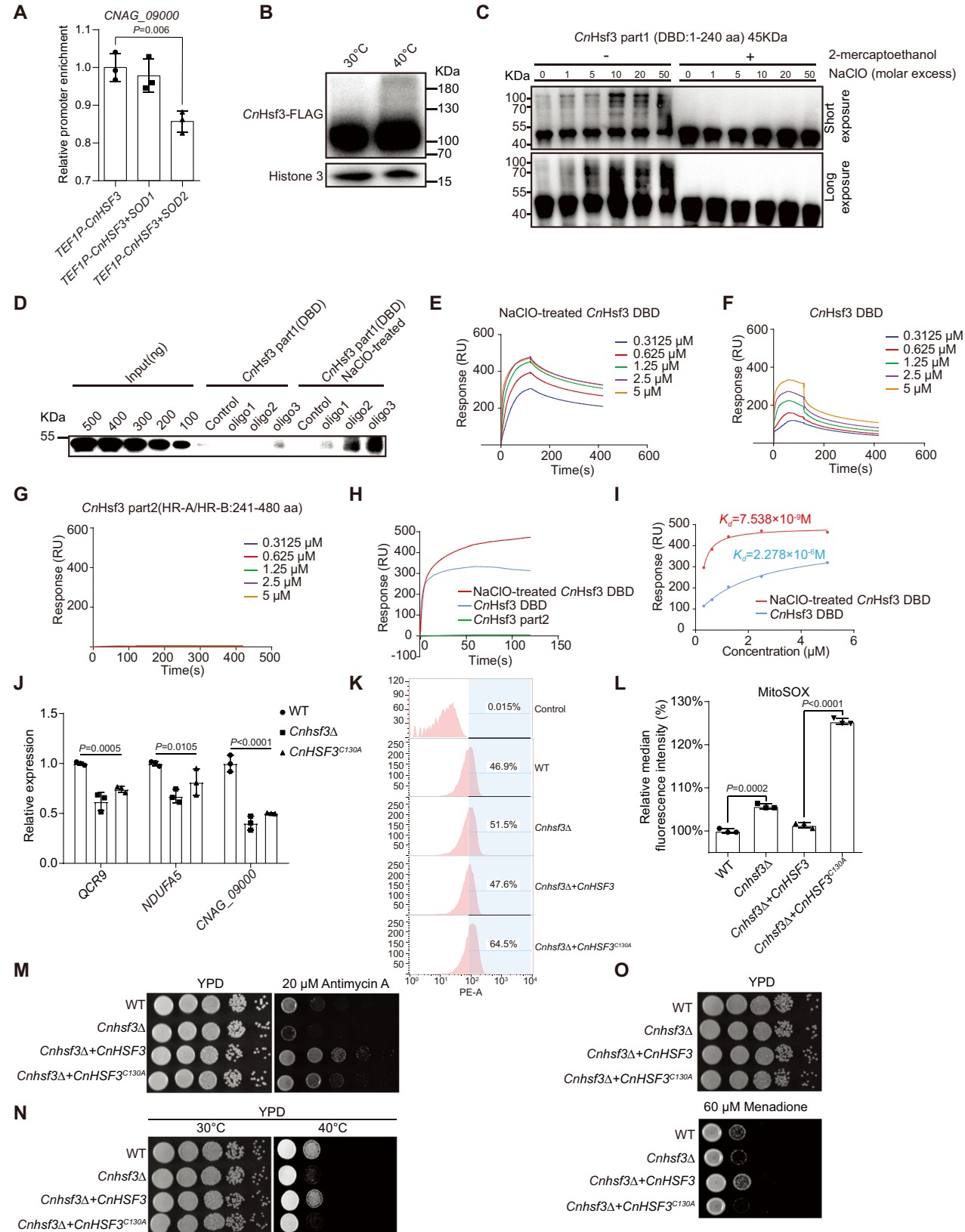

temperatures. Using a bacterial expression system, we expressed and purified the *Cn*Hsf3 DBD and treated it with NaClO, a potent oxidant inducing higher-molecular-weight protein production (Fig. 7C). A protein–DNA co-IP experiment was then carried out using three biotin-labeled *Cn*Hsf3-mitochondrially targeted oligonucleotides (Fig. 7D). Results showed that NaClO-treated *Cn*Hsf3 DBDs readily bind to all

target oligonucleotides, whereas the untreated *Cn*Hsf3 DBD remained unbound or demonstrated limited binding as compared to the NaClO-treated sample. The EMSA analysis demonstrated that the NaClO-treated DBD shifted to give rise to a high molecular weight DNA–protein complex (Supplementary Fig. 9A). Moreover, surface plasmon resonance (SPR) analysis recapitulated the co-IP DNA binding

**Fig. 7 | Oxidization of *Cn*Hsf3$^{C130}$ on the DBD is essential for its function.**
**A** *Cn*Hsf3 ChIP-PCRs in *SOD*-overexpressing strains. *Cn*Hsf3 ChIP-PCR was performed in *SOD1*- and *SOD2*-overexpressing strains (*n* = 3) using *CNAG_09000*. Two-tailed unpaired *t*-tests were used. **B** Immunoblotting of *Cn*Hsf3-FLAG. Proteins were isolated without β-mercaptoethanol, then immunoblotting was performed. Data were obtained from three independent experiments and representative images are shown. **C** Immunoblotting of *Cn*Hsf3-part1-6His. Purified *Cn*Hsf3 part1 was treated with NaClO, then immunoblotting was performed. Data were obtained from two independent experiments and representative images are shown. **D** In vitro assay of the binding of *Cn*Hsf3 DBD to mitochondrial DNA fragments. DBD or NaClO-treated DBD was co-incubated with biotinylated oligonucleotides, then IP with streptavidin magnetic beads was performed followed by immunoblotting. Data were obtained from three independent experiments and representative images are shown.
**E**−**H** SPR analyses of the binding of *Cn*Hsf3 DBD to mitochondrial DNA fragments. Indicated concentrations of *Cn*Hsf3 DBD, NaClO-treated DBD, and non-DBD

(*Cn*Hsf3 part2) were used, then SPR assays were performed using oligonucleotide 3. **I** Calculation of the equilibrium dissociation constant. The equilibrium dissociation constant ($K_d$) was calculated for *Cn*Hsf3 DBD and NaClO-treated DBD. **J** qRT-PCR of ETC genes in the *CnHSF3$^{C130A}$* strain. *CnHSF3$^{C130A}$* cells (*n* = 3) were incubated at 40 °C for 3 h, then the expressions of *QCR9*, *NDUFA5*, and *CNAG_09000* were quantified. One-way ANOVA tests were used. **K** Quantification of mtROSs. Wildtype, *Cnhsf3Δ*, *CnHSF3*, and *CnHSF3$^{C130A}$* strains were loaded with MitoSOX and incubated at 40 °C for 30 min, then flow cytometry was used to evaluate fluorescence and count cells. **L** Quantification of MitoSOX signals. Fluorescence signals of MitoSOX-stained cells (*n* = 3) were quantified and plotted. Two-tailed unpaired *t*-tests were used.
**M**−**O** Spotting assay of the *CnHSF3$^{C130A}$* strain. Indicated strains were spotted onto YPD agar supplemented with antimycin A or menadione, then incubated at 30 or 40 °C for 2 days. Data are expressed as mean ± SD. Source data are provided as a Source Data file.

assay results that showed a binding disassociation constant ($K_d$) of $2.278 \times 10^{-6}$ M for the reduced form and $7.538 \times 10^{-9}$ M for the oxidized form of the *Cn*Hsf3 DBD (Fig. 7E–I).

Finally, to identify potential sites of oxidation, we mutated the single cysteine residue on the *Cn*Hsf3 DBD to alanine and constructed the *CnHSF3$^{C130A}$* strain. This strain demonstrated the normal cellular localization of *Cn*Hsf3 in nuclei and mitochondria (Supplementary Fig. 9B, C) and showed a phenotype compatible with that of *Cnhsf3Δ*, but with repressed ETC gene expression (Fig. 7J), loss of binding to mitochondrial gene promoter (Supplementary Fig. 9D), greater mtROS production (Fig. 7K and L), and growth defects upon mitochondrial stress (Fig. 7M–O and Supplementary Fig. 9E, F). Together, these results demonstrated that mitochondrial oxidative stress triggers mtROS generation and, consequently, the oxidation of the cysteine residue on *Cn*Hsf3 DBD, which in turn enhances the binding ability of *Cn*Hsf3 to target genes that activate the mitochondrial protection mechanism.

## Discussion

Mitochondria govern a wide range of cellular activities, including proliferation, differentiation, and even death, and mitochondrial stress has been linked to various cellular defects and even ailments in humans[1,4]. Consequently, understanding mitochondrial stress responses and deciphering mtROS homeostasis are important for understanding the underlying molecular mechanisms. While previous studies using fungal model organisms are rare, studies in mammals have identified a limited number of mitochondrial targeting transcription factors[36,37]. For example, the tumor suppressor p53 binds to a Bcl-2 family protein, activating mitochondrial apoptosis by inducing mitochondrial outer-membrane permeabilization[38]. The mitochondrial translocation of p53 is also linked to ferroptosis[39]. The transcription factor NK-κB inhibits the expression of mitochondrial RNAs encoding cytochrome *c* oxidase III and cytochrome *b*, consequently blocking mitochondrial respiration[40]. Finally, mammalian Stat3 attenuates electron transfer by directly interacting with GRIM19, a complex I component in mitochondria[41,42]. Despite their mitochondrial-targeting features, these molecules clearly lacked mitochondrial-targeting sequences, leading to challenges in distinguishing their organellar-specific functions[37].

The HSF family proteins represent another type of non-mitochondrial-targeting transcription factor that performs a complex interplay between cytosolic proteostasis and mitochondrial functions. HSF proteins have been shown to be nuclear-regulating transcription factors and master regulators of UPRs, governing proteostasis in both the cytosol and mitochondria[13,29,43]. Intriguingly, our present study challenges this view by showing that *Cn*Hsf3 simultaneously targets both nuclei and mitochondria. Importantly, *Cn*Hsf3 appears to sense and detoxify mtROS independent of UPR.

In fungi, the understanding of the evolutionary and biological functions of the HSF family has been hindered due to limited studies. *S. cerevisiae* and *C. albicans* Hsf1 proteins remain the only HSF proteins identified prior to our studies. We found that most fungal genomes encode at least two HSF members, suggesting their conservation and perhaps important functions. We also found that the *Cn*Hsf3 binding motif partially overlaps with that of other HSFs. A hydrophilic residue substitution occurs at the DNA binding helix of *Cn*Hsf3 in contrast to other Hsf1 proteins, resulting in an adenine-rich motif. We found that the unique DBD feature of *Cn*Hsf3 results in an exclusive interaction with the promoter sequences of protein chaperones leading to its functional divergence, one that departs from UPR regulation in favor of acting as a master regulator for metabolic processes. Studies showed that hHsf1 or hHsf2 form oligomeric complexes upon activation[10,44]. Our EMSA analysis detected multiple molecular weights of protein−DNA complex species, suggesting *Cn*Hsf3 may oligomerize when binding to DNA, and this oligomerization is a protein concentration or oxidation-dependent phenomenon.

*Cn*Hsf3 modulates genes involved in the TCA cycle and mitochondrial ETC but in an opposing fashion. It represses TCA genes while activating mitochondrial ETC genes. Additionally, *Cn*Hsf3 does not bind to the promoters of nuclear-encoded ETC genes, indicating its distinct regulatory mechanisms between the two organelles. *Cn*Hsf3 demonstrates a broad range of regulation on genes encoding complexes I, III, IV, and V, unlike other mitochondrial transcription factors, such as CREB and NK-κB, which bind to the D-loop of the mitochondrial genome and modulate only a fraction of ETC genes[40,45]. Others have demonstrated that Stat3 binds with mtDNA in association with mitochondrial transcription factor A (TFAM)[46]. However, *Cn*Hsf3 does not interact with the TFAM homolog (*CNAG_02115*) (Supplementary Data 5). *Cn*Hsf3 mitochondrial importing mechanism does resemble that of CREB[47]. We identified an Hsp60-like mitochondrial protein chaperone and the Tim complex (Tim44) as *Cn*Hsf3 interacting proteins, revealing a potential importing system for *Cn*Hsf3. Interestingly, while MTS motifs were found in *Cn*Hsf3 and hHsf5, none were identified in other HSFs, including *Sc*Hsf1, *Cn*Hsf1, hHsf1, and hHsf2) (https://ihg.helmholtz-muenchen.de/ihg/mitoprot.html)[48]. We found that modulation of MTS in *Cn*Hsf3 results in its exclusion from mitochondrial localization which also abolishes its interaction with Tim44. Finally, we provided evidence demonstrating that both the NLS and MTS of *Cn*Hsf3 are essential for maintaining mitochondrial function and mtROS homeostasis. It remains to be examined why the *Cn*Hsf3 NLS mutant produced higher levels of mtROS than the wild-type or MTS mutant strains, or whether it implies that the function of nuclear *Cn*Hsf3 is more important in maintaining mtROS homeostasis than does mitochondrial counterpart.

mtROS generation in the *Cnhsf3Δ* cells is likely to be derived from two sources: the activation of TCA cycle genes and the repression of ETC genes. Induction of the TCA cycle turnover rate results in the

accumulation of fuels for the ETC, whose impairment in complexes I and III lead to the generation of intramitochondrial oxidative stress. As demonstrated by others, ROS is generated by a reaction catalyzed by α-ketoglutarate dehydrogenase[49]. We found that the gene encoding α-ketoglutarate dehydrogenase (*KGD2*) in *C. neoformans* was significantly induced in the *Cnhsf3Δ* strain, suggesting a potential function of *KGD2* in ROS generation. Thus, while regulating TCA and ETC, *Cn*Hsf3 also regulates mtROS detoxification by systemically modulating the respiration process.

Mitochondrial homeostasis and integrity are often linked with fungal pathogenicity[50,51]. Previous work in *C. gatti* demonstrated a unique mitochondrial gene expressing profile and an enhanced tubular morphology formation in resistance to host ROS killing[52]. In our study, the *Cnhsf3Δ* strain, which was incapable of maintaining mitochondrial morphology and intramitochondrial ROS levels, demonstrated moderate but consistent attenuation in fungal virulence as evidenced by prolonged survival and reduced lung tissue fungal burden, despite no noticeable growth defect at 37 °C. With the mtROS generation being the critical factor that causes cell growth impairment in the *Cnhsf3Δ* strain, the attenuation in fungal virulence might be more likely to be determined by host pulmonary immunity than the temperature.

Despite that, the morphological alterations detected in the *Cnhsf3Δ* strain at 30 °C, the expression of ETC genes remained unchanged, mainly due to that 30 °C is not a mitochondrial stress inducer. Upon stimulation by mitochondrial stress inducers, *Cn*Hsf3 functions as a universal mitochondrial integrity modulator in *C. neoformans*, protecting cells from multiple stresses, including elevated temperatures, the presence of mitochondrial inhibitors, and even mitochondrial genome damage. *Cn*Hsf3 targets NADH-ubiquinone oxidoreductase subunit (*NDUFA5*) and NADH dehydrogenase subunit (*CNAG_09000*) genes of complex I and the ubiquinol-cytochrome *c* reductase subunit (*QCR9*) gene of complex III that function to detoxify mtROS. Cells overexpressing *NDUFA5* and *QCR9* rescued the heat-sensitive phenotype of the *Cnhsf3Δ* strain. *Cnhsf3Δ* cells overexpressing *NDUFA5* showed wild-type growth in the presence of menadione, while overexpressing *QCR9* moderated the toxicity of antimycin A that inhibits complex III but not menadione resistance. Moreover, all mitochondrial stresses in *Cnhsf3Δ* cells are readily reverted by the expression of *CnSOD2*, strongly demonstrating that *Cn*Hsf3 regulation is a mechanism that parallels the classic mtROS detoxification process by superoxide dismutase.

Finally, mammalian Stat3 employs cysteine oxidization to scavenge mtROS[37]. We showed that excessive mtROS oxidizes *Cn*Hsf3. Activating *Cn*Hsf3 DNA binding fully depends on the oxidation of the 130-cysteine residue, which induces mitochondrial DNA binding affinity by approximately 300-fold. Point mutations at the cysteine residue revealed its essential role in modulating full *Cn*Hsf3 function: it produces a *Cnhsf3Δ* null phenotype, abolishes binding to the promoter of its target gene, and attenuates the expressions of both nuclear and mitochondrial encoding ETC genes. All this evidence demonstrates an unexpected role of an HSF family member and deciphers the molecular mechanism of the mitochondria-targeting *Cn*Hsf3 in modulating mitochondrial stress. Given the importance of *Cn*Hsf3 in governing essential functions in mtROS homeostasis, our findings underscore the importance of a more thorough evaluation of HSF proteins for disease mechanisms in higher organisms and enlighten the potential targets for antifungal therapy.

## Methods

### Yeast growth and strains
Strains were cultured and maintained in yeast extract-peptone-dextrose (YPD) (1% Yeast Extract; 2% Peptone; 2% D-glucose) or yeast nitrogen base (YNB) (0.67% Yeast Nitrogen Base) media. To effect a biolistic transformation, YPD 2% agar with 1 mM sorbitol was used. For the selection of transformants, G418, NAT, and hygromycin B were added to YPD to a final concentration of 200, 100 μg/ml, and 200 units/ml, respectively. Each gene disruption cassette contained a nourseothricin, G418, or hygromycin B selection marker, and amplification was achieved using the primers shown in Supplementary Data 6. *C. neoformans* mutant strains used in this study are shown in Supplementary Data 7. The manipulations of mutant strains are provided in Supplementary Methods.

### Total RNA preparation and quantitative RT-PCR
Wildtype and mutant strains were independently grown to the mid-log phase in YPD at the indicated temperature. Cells were harvested at $1000 \times g$ for 5 min at 4 °C and were washed twice with ice-cold ddH$_2$O. Total RNA was isolated using the Total RNA Kit I (Omega), and cDNA was synthesized using a Reverse Transcript All-in-one Mix (Mona), then the genomic DNA was removed using TURBO DNA-free™ (Invitrogen). Primers for amplifying target genes are shown in Supplementary Data 6. Data were acquired on a CFX96 real-time system (Bio-Rad), and *ACT1* expression was used as a normalization control. The $^{ΔΔ}$Ct method was used to calculate relative gene expression.

### ChIP-seq and ChIP-qPCR
*CnHSF1-FLAG* and *CnHSF3-FLAG* cells were grown separately overnight in YPD media and were subcultured to an OD$_{600}$ of 0.8 (mid-log phase) at 40 °C. To 200-ml conical flasks containing 1.39 ml 37% formaldehyde, a 50-ml of the cell culture was added, and incubation was allowed at room temperature with gentle rocking for 15 min. To stop the crosslinking reaction, 2.7 ml 2.5 M glycine was added, and the mixture was held for 5 min. Cells were harvested at $1000 \times g$ for 5 min at 4 °C and were washed twice with ice-cold PBS containing 125 mM glycine. Chromatin was extracted. The ChIP-seq library was generated using the MicroPlex Library Preparation Kit v2 (Diagenode, Liège, Belgium) according to manufacturer instructions. The ChIP DNA libraries were sequenced using an Illumina HiSeq 2500 Platform by Shanghai Personal Biotechnology Cp. Ltd. Raw reads were mapped to the *C. neoformans* H99 genome (downloaded from http://fungidb.org/fungidb/) using the Bowtie 2 suite (version 4.1.2), as described by Langmead and Salzberg (2012). Visualization of ChIP-seq peaks was performed using IGV (Integrative Genomic Viewer, version 2.3.98). The gene abundance of immunoprecipitation was analyzed using quantitative real-time PCR (CFX96 real-time system; Bio-Rad), employing the specific primer pairs shown in Supplementary Data 6 and IgG was used as the negative control for ChIP-qPCR. The human and *Saccharomyces cerevisiae* heat shock factor motifs were downloaded from the JASPAR website (http://jaspar.genereg.net).

### Immunoblotting and co-immunoprecipitation
Overnight cultures of various *C. neoformans* strains were diluted separately in fresh YPD media and incubated at indicated temperatures to an OD$_{600}$ of 0.8 (mid-log phase). To perform immunoblotting, the fungal cells were resuspended in HEPES buffer (50 mM HEPES, 140 mM NaCl, 1% Tritone X-100, 1 mM EDTA, and protease inhibitors), and samples were processed using a Mini-Beadbeater-16 (BioSpec). Protein immunoprecipitation was performed as described elsewhere (Li YJ et al., 2017). Briefly, cell proteins were extracted using lysis buffer (50 mM Tris−HCl, 150 mM NaCl, and 0.1% NP-40; pH 7.5) with a 1X protease inhibitor cocktail (CWBIO) and 40 mM PMSF. Aliquots of protein extracts were retained as input samples. Three-microgram lysed protein samples were incubated with anti-Flag magnetic beads (MedChemExpress) at 4 °C overnight. The beads were washed three times with TBS buffer (50 mM Tris−HCl, 150 mM NaCl, and 1% Triton X-100; pH 7.4), and the bound proteins were extracted in protein loading buffer with or without 1% 2-mercaptoethanol at 95 °C for 5 min. Protein

samples were separated using 12% SDS–PAGE electrophoresis, transferred onto nitrocellulose membranes, and blocked with 5% milk. Immunoblotting assays were performed using the anti-Flag mouse monoclonal antibody (1:5000 dilution; Transgene), anti-HA (C29F4) rabbit mAb (1:5000 dilution; Cell Signaling Technology), anti-histone H3 (D1H2) XP® rabbit mAb (1:5000 dilution; Cell Signaling Technology), goat anti-mouse IgG (H + L) HRP secondary antibody (1:5000 dilution; Thermo Fisher Scientific), and goat anti-rabbit IgG (H + L) HRP secondary antibody (1:5000 dilution; Thermo Fisher Scientific). The signal was captured using a ChemiDoc XRS + (Bio-Rad).

## Subcellular fractionation and immunoblotting

Mitochondrial protein extractions were performed according to the protocol accompanying the Yeast Mitochondrial Extraction Kit (Beijing Baiaolaibo Technology, HR0257). Briefly, overnight cultures of strains were diluted to an $OD_{600}$ of 0.5 and cultured for 3 h at 30 or 40 °C. The fungal cells were centrifuged at $1000 \times g$ for 5 min and washed twice with pre-cold PBS. Cells were incubated in PBS solution with 0.5% 2-mercaptoethanol for 20 min, then centrifuged at $1000 \times g$ for 10 min at 4 °C. Cell pellets were resuspended with Solution A and were incubated for 15 min at 30 °C, then centrifuged at $1000 \times g$ for 10 min at 4 °C. Pellets were then washed with Washing Solution D. Solution B was added to the pellets, and the two were mixed at a gentle speed for 4 h. The mixture was centrifuged at $2000 \times g$ for 10 min, and the pellets were washed twice with Washing Solution D, after which the pellet was resuspended in Solution C, and the mixture was incubated for 30 min at 4 °C. A portion of the protein extract (100 µl) was retained as total protein. The remaining mixture was centrifuged at $4000 \times g$ for 20 min at 4 °C to remove unbroken cells and nuclei. The supernatant was centrifuged at $12,000 \times g$ for 20 min at 4 °C, and the pellets were resuspended in Preservation Solution. The final solution was centrifuged at $12,000 \times g$ for 20 min at 4 °C, and the mitochondrial fraction protein was collected. The total protein and the mitochondrial fraction protein were analyzed using immunoblotting assays, employing the anti-Flag mouse monoclonal antibody (1:5000 dilution; Transgene), anti-HA (C29F4) rabbit mAb (1:5000 dilution; Cell Signaling Technology), anti-histone H3 (D1H2) XP® rabbit mAb (1:5000 dilution; Cell Signaling Technology), goat anti-mouse IgG (H + L) HRP secondary antibody (1:5000 dilution; Thermo Fisher Scientific), and goat anti-rabbit IgG (H + L) HRP secondary antibody (1:5000 dilution; Thermo Fisher Scientific). The signal was captured using a ChemiDoc XRS + (Bio-Rad).

## Mass spectrometry of *Cn*Hsf3-interacting proteins

The *CnHSF3-FLAG* strain was grown in YPD medium overnight, then inoculated into 50 ml fresh YPD medium at an $OD_{600}$ of 0.2 and shaken in an orbital shaker until $OD_{600}$ reached 0.8. As a control group, the *CnHSF3* deletion strain was treated in the same way. The protein was extracted as described in the co-IP protocol. Aliquots of protein extracts were retained as input samples. Three-microgram lysed protein samples were incubated with Anti-Flag Magnetic Beads (MedChemExpress) at 4 °C overnight. The beads were washed with TBS buffer (50 mM Tris–HCl, 150 mM NaCl, 1% Triton X-100, pH 7.4) three times. The final peptide mixture is obtained by in-gel enzymatic hydrolysis and solid phase extraction. Orbitrap Q-Exactive-plus was used to acquire the Mass spectrometry data. The obtained mass spectrum data was retrieved by Mascot (version 2.5.1). The database used in this project is UniProt database: uniprot_cryptococcus_neoformans_grubii_serotype_A_7441_20180905.fasta. Data quality control is processed by Scaffold Q+ (version 4.6.2). The quality control parameters are: false discovery rate (FDR) of protein <1.0%, false discovery rate (FDR) of peptide <1.0%, and at least one specific peptide is identified for each protein. Quantitative calculations were performed using Maxquant (1.5.2.8) software. The quantitative value in each sample (iBAQ) was used for the Student's *t*-test.

## Analysis of protein–DNA binding

A biotinylated double-stranded DNA probe was formed by heating 400 pmol biotinylated oligonucleotide pairs, shown in Supplementary Data 6, to 95 °C. Separately, 3 µg purified *Cn*Hsf3-part1 protein was added to a 50-fold molar excess of NaClO and held for 60 min at room temperature. The protein expression and purification procedures are provided in the Supplementary Methods. The biotinylated double-stranded DNA probe was incubated with 40 µl Streptavidin Magnetic Beads (MedChemExpress) and mixed with the protein preparation. The mixture was incubated on a rocking platform at room temperature for 2 h with gentle agitation. The beads were washed three times with Buffer I (10 mM Tris–HCl [pH 7.5], 1 mM EDTA, 1 M NaCl, and 0.05% Tween-20), then resuspend in 40 µl protein loading buffer and incubated for 5 min at 95 °C. Protein samples were separated using 12% SDS–PAGE electrophoresis, transferred onto a nitrocellulose membrane, and blocked with 5% milk. Immunoblotting assays were performed using anti-His mouse monoclonal antibody (1:5000 dilution; Sigma) and goat anti-mouse IgG (H + L) HRP secondary antibody (1:5000 dilution; Thermo Fisher Scientific) antibodies, and the signal was captured using a ChemiDoc XRS + (Bio-Rad). The Electrophoretic mobility shift assay (EMSA) was performed as described in the Supplementary Methods.

## Surface plasmon resonance analysis

SPR experiments were performed using a Biacore T200 instrument (Becton Dickinson and Company). In the SPR assay, the 5′ biotinylated oligo (forward strand of oligo 3; GXD798) and non-biotinylated oligo (reverse strand of oligo 3; GXD799), derived from the mitochondrial genome, were prepared and diluted to 0.125 µg/ml in PBS buffer, then heated to 95 °C followed by cooling to room temperature to allow the formation of the double-stranded DNA probe. This probe was used as the ligand and was immobilized on the surface of the SA chip (GE BR-1005-31) by streptavidin, obtaining a binding capacity of 188 RU. Purified *Cn*Hsf3-part1, *Cn*Hsf3-part2, or NaClO-treated *Cn*Hsf3-part1 proteins were diluted in PBS buffer at concentrations ranging from 0.3125 to 5 µM, then passed over the SA chip for 300 s. A flow rate of 30 µl/min was used for the running buffer and sample proteins. The SPR data were analyzed using Biacore T200 Evaluation Software (Cytiva).

## Phylogenetic and protein domain analysis

Protein domain analysis was performed using InterPro (http://www.ebi.ac.uk/interpro/). HSF-like proteins were retrieved from the NCBI protein database using BLASTP with *Saccharomyces cerevisiae* Hsf1. Sequences from orthologous clades were aligned and constructed using ClustalW in the MEGA 5 software package. Sequence alignment and conservation sites were calculated using Clustal Omega (https://www.ebi.ac.uk/Tools/msa/clustalo/). Phylogenetic trees were constructed using the neighbor-joining method and modified using Fig-Tree software (http://tree.bio.ed.ac.uk/software/figtree/).

## Staining

To visualize mitochondria, wildtype and *Cnhsf3Δ* strains were cultured overnight, then subcultured to an $OD_{600}$ of 0.8. Cells were washed twice with PBS, then transferred to preheated Eppendorf tubes containing fresh YPD medium and a fluorescent dye (Mito-Tracker Red CMXRos mitochondrial red fluorescent probe; Beyotime). Strains were washed three times with PBS, then fluorescence images were taken using a Leica TCS SP8 (Leica). Mitochondrial morphology (fragmented, tubular, or diffuse) was determined by counting and classifying the mitochondria from at least 150 cells. To analyze mitochondrial network morphology, the MiNA toolset in ImageJ was employed according to the protocol described elsewhere[53]. The mitochondrial morphology included individuals and networks from 10 cells in each group, and statistically significant differences between groups were determined using the Student's *t*-test.

To detect reactive oxygen species (ROSs), wildtype and *Cnhsf3Δ* strains were separately grown overnight in fresh YPD medium until an $OD_{600}$ of 0.8 was reached. DCFH-DA (Beyotime) and MitoSOX™ Red (Invitrogen) were used as ROS-staining dyes, where the former labeled intracellular ROSs, and the latter, mitochondrial ROSs. Fresh liquid YPD medium supplemented with DCFH-DA or MitoSOX™ Red according to the instructions were mixed with cells that were spun down from 1 ml of either culture. Cells were further incubated at 30 or 40 °C for 30 min, then pelleted using centrifugation and washed three times with PBS. To perform flow cytometry to find the number of cells with ROSs, 1 ml stained cells and 1 ml unstained cells were first diluted with PBS. Flow cytometry data were acquired using a BD LSRFortessa Cell Analyzer, and analysis was performed using FlowJo version 10.0.7r2.

### NADH and ATP quantification

To assay complex I activity, wildtype and *Cnhsf3Δ* strains were cultured separately overnight and grown to an $OD_{600}$ of 0.8, then 100 μl of the resultant culture was mixed with 20 μl CellTiter 96® AQueous One Solution Reagent (Promega), which contains a novel tetrazolium compound, MTS. The mixture was placed in a 96-well assay plate and incubated at 30 or 40 °C for 1 h. NADH levels were determined by measuring absorbance at 490 nm using a MultiskanGO microplate reader (Thermo). All data were normalized using the number of colony-forming units (CFUs).

To monitor the ATP level, cells were prepared as described above. Strains were incubated at the indicated temperature for 30 min, then were disrupted using glass beads and a Bio-Spec bead beater for 3 rounds of 55 s each, resting on ice for 1 min between rounds. ATP levels were measured using the CellTiter-Glo Luminescent Cell Viability Assay (Beyotime) according to manufacturer instructions. The luminescent signal was measured using a Synergy H4 microplate reader (BioTek). All data were normalized to the protein concentration in the sample, which was determined using a MultiskanGO microplate reader (Thermo). Three independent experiments were performed, each including four technical replicates, and a representative data set is presented.

### Ethical statements and animal studies

All animal experiments were reviewed and ethically approved by the Research Ethics Committees of the National Clinical Research Center for Laboratory Medicine of the First Affiliated Hospital of China Medical University (KT2022284). All animal experiments were carried out in accordance with the regulation in the Guide for the Care and Use of Laboratory Animals issued by the Ministry of Science and Technology of the People's Republic of China. Mice were cared with an alternating 12 h light−dark cycle and unlimited food and water supply. Infection experiments were performed in BALB/c mice as described previously with modifications[54,55]. Briefly, 6- to 8-week-old female BALB/c mice were purchased from Changsheng Biotech (China) for animal survival assay. WT, *Cnhsf3Δ*, and *CnHSF3* strains were grown in YPD broth overnight at 30 °C, washed in phosphate-buffered saline (PBS), and resuspended in PBS buffer. Ten mice were anesthetized and intranasally infected by $10^5$ fungal cells in 50 μl of PBS buffer, respectively. The mice were monitored twice daily for signs of infection and humanly killed when endpoints were reached. Mice significant in survival assays were determined using Log-rank (Mantel−Cox) test in GraphPad Prism 6.0. For CFU analyses, *C. neoformans* infected mice were sacrificed at 14-day post-infection, and lung and brain tissues were isolated, weighed, homogenized and plated onto YPD agar plates at 30 °C for 2 days. Colonies were counted and CFUs were calculated. For histopathology analyses, lung tissues were isolated from 14-day-infected mice and fixed in paraformaldehyde. Frozen tissue slides were prepared using a cryostat microtome (CM1860, Leica). Tissue sections of 10 μm in thickness were stained with Periodic Acid-Schiff (PAS) staining[56], and visualized using a ×10 lens (DFC450, Leica).

### Long PCR for mtDNA integrity

To amplify one-third of the mitochondrial genome (6 kb), PrimeSTAR® Max DNA Polymerase (Takara) was used. The long PCR was carried out using 100 ng total DNA with the primers shown in Supplementary Data 6. Amplification conditions were as follows: 95 °C for 2 min followed by 39 cycles of denaturing (95 °C for 15 s), annealing (55 °C for 10 s), and extension (72 °C for 40 s). The PCR products were separated on 0.8% agarose gels.

### Quantification and statistical analyses

All statistical analyses were performed using GraphPad Prism software (GraphPad). Statistically significant differences between the two groups were determined using an unpaired two-tailed Student's *t*-test or two-tailed Mann−Whitney tests. Statistical analyses across two or more groups were performed using one-way ANOVA or two-way ANOVA. Significant changes were found when $p < 0.05$. In transcriptome analyses (detailed transcriptome protocol and data analyses are provided in the Supplementary Methods), differentially expressed genes were defined as those with fold changes exceeding 1.5 or <0.667 in addition to $p < 0.05$. The ChIP-seq peaks were analyzed using the MEME Suite (Motif-based sequence analysis tools, version 5.3.3) and a $p$-value cutoff of $10^{-4}$. In the mass spectrometry of *Cn*Hsf3-interacting proteins, significantly different proteins were found when the ratio was >1.5 and $p < 0.05$.

### Reporting summary

Further information on research design is available in the Nature Research Reporting Summary linked to this article.

## Data availability

The transcriptome (RNA-seq) and ChIP-seq data are deposited in NCBI's Gene Expression Omnibus (GEO) and can be accessed through GEO Series accession ID GEO: GSE183184. Metabolomics data (performed as described in the Supplementary Methods) have been deposited to the EMBL-EBI MetaboLights database with the identifier MTBLS5745. The mass spectrometry proteomics data have been deposited to the ProteomeXchange Consortium via the PRIDE partner repository with the dataset identifier PXD033799. Source data are provided with this paper.

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

## Acknowledgements

We thank Prof. Ren Sheng at Northeastern University for the technical assistance with the SPR experiment. We thank Profs. Dennis Thiele, Xiaorong Lin, Ren Sheng, and Yong-Qiang Fan for critical comments on the manuscript. Funds for this program were provided by the National Natural Science Foundation of China (31870140 to C.D.), the Fundamental Research Funds for Central Universities of China (N142005001 and N172002001 to C.D.), and Liaoning Revitalization Talents Program (XLYC1807001 to C.D.). Research in P.W. lab was supported by the National Institutes of Health (US) awards AI156254 and AI168867.

## Author contributions

C.D., K.Y., J.L., and X.G. designed the project. X.G., Y.F., and Shengyi S. conducted experiments. X.G. and H.L. carried out bioinformatic analyses. Y.L., W.D., and C.S. helped with molecular biology. C.L., Y.G., Y.M., and Y.N. contributed to strain generation. T.S. performed protein mass spectrometric analysis. T.G., S.Y., and T.L. carried out flow cytometry and SPR analysis. C.D., Sixiang S., and X.G. participated in data analysis. C.D., P.W., J.L., and X.G. wrote the manuscript. All authors reviewed and edited the manuscript.

## Competing interests

The authors declare no competing interests.
