## [Peer Review File · Nature Communications]

Cryptococcal Hsf3 controls intramitochondrial ROS homeostasis via regulating the respiratory processREVIEWER COMMENTS

Reviewer #1 (Remarks to the Author):

In the manuscript "Cryptococcal Hsf2 governs mitochondrial ROS homeostasis by regulating respiration independent of UPR", the authors identify a new role for a *C. neoformans* HSF family member that participates in regulation of mitochondrial gene expression in response to oxidative stress. The authors provide compelling evidence that loss of Hsf2 sensitizes *C. neoformans* to temperature and oxidative stress, and is required for mitochondrial-dependent carbon source utilization at elevated temperature. This conclusion is supported by transcriptomic, metabolomics and IP-MS. Interestingly, Hsf2 binds an A-rich consensus sequence, and the DNA binding activity appears to be dependent on the oxidation state of a cysteine in the DNA binding domain, making Hsf2 a sensor of mitochondrial redox balance. The experiments are well-designed and, in general, the data support the conclusion. Below are a few weaknesses that should be addressed.

1. Oxidation of Hsf2 leads to an increase in MW beyond that which would be expected by oxidation, suggesting potential dimerization or aggregation. This data, in light of the fairly low resolution MEME logo for Hsf2, may suggest that DNA binding is multimer dependent. Perhaps analysis of the motif as two motifs (two, 6-base motifs) would yield a stronger consensus? Further, DNA IPs are presented, but EMSAs might help to determine if DNA binding is multimer dependent. If so, the effect of oxidation on multimerization may be warranted. There is data suggesting mammalian HSF1 and HSF3 bind as trimers.
2. There's approximately a 10% increase in mtROS in Hsf2 consistently across the time course presented in Figure 3H and I, but a 40% increase in cytoplasmic ROS. This is interesting, and it would be important to understand if cytoplasm ROS is unchanged as mtROS in control experiments as in Fig 3H. The authors state that the cytoROS accumulation is caused by mtROS accumulation, but the data as presented don't necessarily support this. Also, is 10% increase in mtROS similar to that seen by treatment with an uncoupling agent like antimycin A? It would be nice to see a positive control for mtROS induction and its effect on cytoROS to appreciate where this hsf2 effect is on magnitude scale.
3. The authors conclude that the HSF2C130A mutant is defective in DNA binding, but in light of comment 1, DNA binding should be tested directly either by ChIP, IP or EMSA, or SPR, all of which are shown for the DBD of Hsf2. It would be ideal to determine the contribution of direct anti-oxidant function apart from transcriptional activation to the hsf2 mutant phenotype. Is there a mutation that would abrogate DNA binding, but that would leave the oxidant-sensitive cysteine intact?
4. The authors conclude electron leakage from the etc is the source of ROS. It would be helpful to identify the specific results that suggest this. It is unclear which data support this conclusion.
5. The authors talk a lot about MQC, but the direct link between the reported data and how they support a defect in MQC is unclear. This could be made more apparent in the results.
6. The authors use only parametric statistical tests, including for the ordinal data that is scoring mitochondrial morphology. They should consider non-parametric tests for these data.
7. The authors include "independent of the UPR" in the title. I think this detracts from the fundamental role of Hsf2 in redox sensing and mt function.

Reviewer #2 (Remarks to the Author):

In this manuscript, Gao et al. investigated the function of Hsf-like protein 2 (Hsf2) in mitochondrial quality control and homeostasis in *C. neoformans*. They found that expression of HSF1 is highly upregulated in response to high temperature (40°C) and Hsf2 is involved in thermotolerance of *C. neoformans* at 40°C. Interestingly, the ChIP-seq analysis indicate that Hsf2 binds to the promoters of genes involved in metabolism, but not in protein chaperones, suggesting that Hsf2 regulates heat shock response via UPR-independent manner. Subsequently, the authors showed that Hsf2 is required for mitochondrial metabolism and responding to mitochondrial reactive oxygen species (mtROS). Notably, the authors found that Hsf2 is both mitochondrial and nuclear targeting transcription factors and activate transcription of genes involved in electron transport chain to detoxify

ROS. They also showed that Hsf2 is a sensor and modulator for mtROS homeostasis under various conditions that lead to mitochondrial dysfunction. Finally, the authors demonstrated that oxidation of cysteine residues in its DNA binding domain reduce Hsf2 binding to the mitochondrial genome.

Overall the authors provided a large amount of molecular genetic and biochemical data for the novel roles of Hsf2 in mitochondrial quality control and homeostasis in *C. neoformans*. The data presented by this study could be very informative and useful for those working in this pathogen. Having said that, I have the following major criticism, which makes me less enthusiastic for this work. The authors did not provide any *in vivo* data regarding the functions of Hsf2 in the pathogenicity of *C. neoformans*. If Hsf2 is a universal mitochondrial integrity modulator in *C. neoformans*, does inhibition of Hsf2 actually impact the pathogenicity of the pathogen? Previously, the Robin May group demonstrated that *Cryptococcus gattii* adopts a mitochondrial tubularization in response to host ROS and these fungal cells facilitate the rapid growth of neighboring *C. gattii* cells with non-tubular mitochondria, leading to increased pathogenesis of infected *C. gattii* population (Voelz et al. 2014 Nat Commun). In this manuscript, the authors mentioned that deletion of HSF2 resulted in dysregulation of mitochondrial morphology in *C. neoformans*. Therefore, although *C. neoformans* and *C. gattii* are two different *Cryptococcus* species, this is one of the issues that the authors should think about for the potential roles of Hsf2 in the pathogenicity of *C. neoformans*. Collectively, the lack of data showing the role of Hsf2 in the pathogenicity of *C. neoformans* markedly reduces the significance of this work.

I have the following additional comments and concerns for this work.

1. The authors did not appear to properly cite previously published papers and discuss them, which are relevant for this work. For example, here the authors mentioned that they identified three Hsf1-like proteins and named Hsf1, Hsf2 (CNAG_04036), and Hsf3 (CNAG_04176). However, these three Hsf1-like proteins in *C. neoformans* have been already reported in the previous papers (Jung et al. 2015 Nat Commun; Yang 2017 Genetics). In these previously published papers, the three Hsf proteins were named as follows: Hsf1 (CNAG_07460), Hsf2 (CNAG_04176), and Hsf3 (CNAG_04036). Therefore, as these previous publications exist, the authors should follow their gene names to avoid any potential confusion among readers. In addition, Yang and colleagues previously reported the essentiality of Hsf1 in the growth of *C. neoformans* by conditional promoter replacement and sporulation analysis of heterozygous HSF1/hsf1 mutants (Yang 2017 Genetics). They also showed that expression of HSF1 is downregulated, which is in agreement with the data presented by the current work, and overexpression of HSF1 increases the thermotolerance of *C. neoformans*. Yang et al. also demonstrated that Hsf1 can physically bind to the promoter of SSA1 through CHIP-PCR analysis, which is in agreement with the data presented by the current work.

2. Here the authors tested the thermosensitivity of the hsf2 mutant at 40°C, which is not the host biological temperature. What is the role of Hsf2 in the growth of *C. neoformans* at 37°C?

Reviewer #3 (Remarks to the Author):

General comments:

In this manuscript, the authors have identified and characterized CnHsf2, an HSF-like protein or a protein possessing domains related with HSF DNA-binding domain in *Cryptococcus neoformans*. Several vertebrate HSF-like proteins have been identified and characterized, but their roles are not well known. The authors first determined that CnHsf2 regulates expression of nuclear-encoded genes involved in the TCA cycle and mitochondria-encoded electron transfer chain genes, and protects cells from heat shock and mitochondria oxidative stresses. Localization of CnHsf2 in both the nucleus and mitochondria and its roles in regulation of both nuclear-encoded and mitochondria-encoded genes under stress conditions are unusual. These exciting conclusions are derived from a lot of experiments with high quality. There are several points below that should be clarified before publication.

Specific comments:

1) The authors imply that CnHsf2 is an ortholog of human HSF-like protein, named HSF5, based on the phylogenetic analysis (Fig. S1A). This point is interesting and important because this study first show roles of HSF-like proteins in a cell. In the introduction (lines 81-83), the authors define HSFs as transcription factors that bind to the HSE. Indeed, HSF1 to HSF4 bind to the HSE, but the HSE is not bound by HSF-like proteins, such as human HSF1, HSF2, HSF3 and HSF4, which possess domains related to the DNA binding domain of HSFs. Therefore, an explanation should be required to distinguish the HSF-like proteins from HSFs in the introduction.

2) It is important to show detailed informations to determine whether CnHsf2 is an ortholog of human HSF5 or not. First, the authors should show an alignment of the full-length amino acid sequences of human HSF5 and CnHsf2 in Fig. S1. Second, they should examine in vitro binding of CnHsf2 to a consensus heat shock response element (HSE) and the adenine-rich sequences shown by ChIP-seq data in Fig. 1E by performing EMSA. It is also important to show whether CnHsf2 binding to these sequences and its oligomerization is induced during heat shock. Third, it would be better to determine whether human HSF5 is able to substitute CnHsf2 for target gene expression or some phenotypes.

3) The authors claim that “Hsf2 has the GAA-like motif enrichment in adenine”. This statement may mislead readers about its binding sequences. It would be better to mention that the sequences are adenine-rich, but is different form repeats of nGAAn motif.

4) The authors claim that CnHsf2 localizes in the mitochondria and nucleus in Figs. 4 and 5. However, it seems by western blot data in Fig. 4C that only a little fraction of CnHsf2 localizes in the mitochondria. They should compare CnHsf2 levels in the nucleus, cytoplasm, and mitochondria in the same blot, and should determine whether CnHsf2 moves to the nucleus and mitochondria during heat shock or not. In addition, immunofluorescence data in Fig. 5D clearly showed exclusion of CnHsf2-NLS_mut-GFP. They should also show localization of CnHsf2-MTS_mut-GFP. If this construct is too large to translocate to the mitochondria, they can stain overexpressed CnHsf2-MTS-FLAG and CnHsf2-MTS_mut-FLAG in cells.

5) The CnHsf2 mRNA level was induced during heat shock (Fig. 1A). However, it is unclear what “activation of CnHsf2” means (P10, line 283; P11, line 316). Is CnHsf2 protein level upregulated during heat shock? Is DNA-binding activity (to nuclear-encoded genes) elevated during heat shock? Does oligomeric form of CnHsf2 change during heat shock? The authors should mention these points by showing experimental data.

6) In Fig. 1F, did the authors analyze unstressed cells or heat-shocked cells? Explain this point in the legend.

7) It would be better to re-consider the name “Hsf2”. This is not related with HSF2 in vertebrates, but could be “CnHsf5”. CnHsf1 is an ortholog of vertebrate HSF1, but CnHsf2 and CnHsf3 (in this manuscript) are not orthologs of vertebrate HSF2 and HSF3. Readers in the wide fields (for nature communications) may be confused with these names. One possibility is that Hsf2 and Hsf3 could be HSF-like protein 1 (Hsf1) and HSF-like protein 2 (Hsf2).

Reviewer #4 (Remarks to the Author):

In the manuscript by Gao et al., the authors characterized a new HSF protein (Hsf2) in the pathogen *C. neoformans* that is homolog to human HSF. They conduct experiments in the presence of Heat Shock (HS) or various oxidants and demonstrated that cnHsf2 reacts to the accumulation of ROS which activates the localization of cnHsf2 into the nucleus and mitochondria where it regulates the expression of genes related to the TCA and ETC. Deletion of Hsf2 is viable but causes growth defects in the presence of stress which the authors attributed to a defect in the regulation of mitochondrial quality.

While this is an interesting study that provided new insights into the Heat Shock Factors field, the study is very limited to the characterization of *cnHsf2*. The most important concern is that the study lacks significance in understanding how this mutation, if relevant, influences pathogenesis by this fungus. In other words, is this protein relevant in any way for the pathophysiology of *C.n*? Without biological significance, the scope of the study is very limited.

Major concerns:

As of general comments, several experiments lack important controls, in several cases samples are compared inappropriately and conclusions are not always supported by their results. The authors need to (a) provide individual data points for all graphs and indicate the statistical analyses performed in the figure legend of each figure. (b) All spot assays must be accompanied by growth curves where replicas and statistical analyses can be performed to better quantitate the impact of genetic mutations and/or treatments.

-In Fig. 1B, the authors stated "the loss of HSF2, but not HSF3, attenuates growth at 40°C (Figure 1C), suggesting Hsf2 and Hsf3 have distinct roles as HSR. The effect on growth attenuation between Hsf2D and Hsf3D is very modest. Since the phenotype of Hsf2 seems to be critical for concluding that Hsf2 participates in HSR, the authors should conduct an experiment where such phenotype can be quantitated. A growth curve comparing 30C vs 40C should be shown to indicate the extent of the attenuation and the differences between Hsf2 and Hsf3. Please include a Hsf1 deficient mutant (in the case that Hsf1D is not viable) in the analysis as a reference (positive control), since the comparison with the 'main HSF' is necessary to draw conclusions about how important Hsf2 is in the HSR.

-Regarding the structural comparisons shown in Fig.1, The comparisons presented seem random, with no description of whether there is any similarity with *C.n* Hsf1. It is unclear how the authors hypothesize that the mechanism of Hsf2 response might be similar to that of other HSFs. The authors should comment on (1) the structural homology between *C.n* Hsf1 and Hsf2, (2) whether this comparison resembles the differences between mammalian Hsf1 and Hsf2, (3) How different/similar *C.n* Hsf2 is compared to other HSFs in yeast and/or mammals.

-The authors conclude that Hsf2 regulates the HSR through mechanisms different from the UPR. Please include a comparison with Hsf1 as a positive control to compare the response of Hsf2D with the HSR expected from a canonical HSF. Please include a quantitative representation of example genes for Hsf2 as shown in Fig. 1I for Hsf1.

-Fig. 2A Please explain how the KEGGs are nearly identical between wt and Hsf2D cells, especially for those genes related to carbon metabolism. Shouldn't the authors expect differences in the pathways that are supposed to be Hsf2 dependent?

-The authors need to better explain why there is no growth phenotype in Hsf2D cells grown in YNB. They indicated that 'Cells grown in the absence of six-carbon sugars were partially rescued at 40C'. but it seems that there is no difference in growth at all and that any growth defects are completely abolished in the absence of sugar. Growth curves should be shown to better determine the extent of the attenuation or amelioration in growth effects by carbon source and should be compared with a Hsf1 deficient mutant.

-In fig. 3C the authors indicated "we employed random hexamer qRT-PCRs to measure mitochondrial encoding ETC gene expression." To justify that they did not find any mitochondrial gene in their RNA-seq studies. However, to the knowledge of this reviewer, there are no reports in the literature indicating that oligo(dT) for reverse transcription could result in the insufficient synthesis of mitochondrial cDNA. The authors claim they are analyzing mitochondrial encoding ETC gene expression just by using hexamers to synthesize cDNA from whole cell extracts. However, it is unclear how the authors distinguish between nuclear encoded and mitochondrial encoded genes just by changing from oligodT to hexamer amplification. Nowhere in the methods section it is described such distinction. Unless the authors purify mitochondria and isolate RNA from mitochondria it would be extremely difficult to distinguish between the RNAs that are coming from nucleus or mitochondria when preparing cDNA from whole cell extracts using either oligodT or hexamers.

-The conclusion that 'six of seven tested ETC genes are positively regulated by Hsf2' is not accurate. The authors only compared wt vs Hsf2D at 40C and what they observed is that those 6 genes are partially repressed in Hsf2D compared to wt. However, some of those genes seem to also be

repressed in WT at 40C compared to 30C or not changed indicating that the response observed in those genes does not correspond to a positive regulation by HSF2. The authors need to statistically compare all conditions to better determine which genes respond to Heat shock (HS) and whether they are regulated by Hsf2.

-Fig. 3D and Fig. 6E. Data representation is confusing. Using IgG negative controls is a standard in ChIP experiments. The authors should include IgG negative controls and data should be relativized to IgG. Then the authors can normalize their data based on % input but a negative IgG control should be used. There is no way to determine if ChIP signals are specific to the antibody without a negative control. Gene CNAG_09004 is not tested in Fig. 3C. Please provide an explanation why this gene was chosen.

-Fig. 3E. Hsf2D+QCR9 should be in the same plate as Hsf2D+QCR9+NDUFA5 to compare on the same plate the benefit of co-expressing with NDUFA5. Since spots are not on the same plate it is impossible to compare whether co-expressing QCR9 with NDUFA or CNAG_09000 have similar or different effects. The authors need to provide growth curves to better quantitate (including statistics) the impact on growth of the different ETC expressing genes. In addition, since co-expression with only nuclear encoded genes (based on the authors conclusions) seems to restore growth deficits, then the impact on mitochondria ETC by Hsf2 does not seem to play a key role in the temperature sensitive phenotype.

-Fig. 3M. Without growth curves and cultures with and without NAC simultaneously compared is difficult to assess whether there is a difference in growth in Hsf2D cells at 40C treated with NAC or not.

-The authors claim that 'Hsf2 responds to ROS overload of the mitochondrial matrix rather than an elevated temperature.' The authors do not provide data to support this conclusion. They only showed alterations in ETC gene expression and ROS when comparing Hsf2D at 40C. To claim that Hsf2 does not respond to elevated temperature (independent of ROS) the authors should show that no changes in HSF2 genome wide binding patterns occurs when conducting HSF2-ChIP-experiments in the presence of HS + antioxidants.

-Fig. 4C. The authors need to load 30C and 40C samples in the same gel to compare whether HS alters the amount of Hsf2 found in the mitoc. This seems to be important to understand the mechanism by which Hsf2 regulates mitoc stability under HS. The authors did not show anywhere in the manuscript that HS or any other stress increases the localization of Hsf2 in the mitoc (as it would be expected).

-Fig. 4E. This experiment should have been carried out in the presence of wt HSF2-FLAG and loaded in the same gel to compare side by side the impact of mutating the MTS. Without this important control the reviewer cannot see whether this mutation prevents Hsf2 from getting in the mitoc.

-Fig. 4F. should also include wt Hsf2-Flag as a positive control. Not showing a band in the HA blot does not necessarily mean there is no interaction since absence/presence of signal is relative to the time of exposure. Without a reference to demonstrate that the relative difference with a positive control is difficult to assess whether the MTS mutation does impact or not binding.

-Fig. 5B. Hsf2-NLSmut should be included in this experiment to demonstrate mitochondrial localization and to be compared to Hsf2-MTSMut within the same gel. Same applies to Fig. 5C. In addition, samples from 30C and 40C in Fig. 5B should be included in the same gel and imaged simultaneously to determine whether there are temperature-dependent differences in the localization.

-Fig. 5D, Similarly to the previous panels, the authors should have included Hsf2-MTSMut as a control to show nuclear localization and to be compared with Hsf2-NLSmut.

-The authors indicated that 'binding of Hsf2 to mitochondrial DNA is proportional to mtROS production'. Please provide quantitative data to support this conclusion. No experiments or analyses addressing the proportionality of this effect is shown.

-In Figure 7, the authors conclude that oxidation of Cys130 is necessary for the mitochondrial protection by Hsf2 since a Hsf2C130A mutant behaves similarly to hsf2D. The authors need to show that this mutation does only alter DNA binding and not protein localization. It is possible that the Cys130 is a critical residue within the Hsf2 structure to mediate compartment localization and therefore its mutation will generate a protein that is not functional.

Minor concerns:

- The authors could consider referencing to Hsf2 throughout the manuscript as cnHsf2 since the nomenclature can be confused with mammalian Hsf2.
- The description of data presented in Fig. 1B is not accurate: the authors indicated “expression of both HSF1 and HSF2 is responsive to elevated temperatures (Figure 1B)”, The expression of Hsf3 is also ‘responsive to elevated temperature. Indeed, Hsf2 and Hsf3 have a similar response to 40C while HSF1 presented an opposite response. The authors should better discuss these differences among the three HSFs.
- Define the composition of YPD and the difference with YNB in the methods section, this is relevant to understand the discrepancy in the grown effects between these media and the experiments where different carbon sources are added.
- The representation in Fig.2D is difficult to follow. Please provide a representative result where comparisons can be made quantitatively.
- In Fig. 5E, Is this experiment conducted at 30C? Provide info in the figure legend. The authors showed a significant reduction in the expression of CNAG_09000 between wt and Hsf2D of ~50%. However, this contradicts the data presented in Fig. 3C where the two graph bars (no statistical data provided) seem to be identical.
- In Fig. 5H the authors indicated “mtROS-producing cell counts and signal intensities of these mutants were significantly greater than those of the wild-type cells (Figures 5G and 5H). This sentence is not accurate. The mutants present higher mtROS compared to Hsf2D cells complemented with HSF2. The authors should consider including statistical comparisons between the mutants and Hsf2D cells too. When doing such comparison, it can be seen that MTSmut produced similar ROS than Hsf2D but NLSmut is worst, which would suggest that the localization in the nucleus plays a more important role than the localization in the mitoc.
- Concentrations for menadione/antimycin ‘ μ m’ should be μ M.
- Fig.6G. showed spots cut out from the plate. Please provide all spots on the same plate so they can be compared with each other.
- It would be important to comment whether there is a MTS is other yeast and/or mammalian HSFs? Is this unique to C.n. Hsf2?
- Considering the alterations observed in the mitoc morphology in Hsf2D cells even at 30C, it would be expected to see alterations in gene expression of mitoc related genes. However, as shown in Fig. 3C, mitoc ETC genes does not seem to show differences between WT and Hsf2D cells at 30C. The authors should discuss this discrepancy.
- The authors indicated “Menadione triggers enrichment of Hsf2 in the mitochondrial genome, and bindings at CNAG_09000 and CNAG_09009 were ameliorated in a time-dependent manner’. However, Fig. 6I showed a time-dependent increase in Hsf2 binding on those genes.
- Fig. 1A, and 1F provide a numeric range for the color scale
- Fig. 1B, add the ACT1 normalization control in the figure legend.
- Fig. 1H. in the legend the authors indicated “ChIP-seq data of the protein chaperones MDH1 and HXK1 are shown”. These genes are not chaperones.
- Fig. 1I. Provide information about how enrichment is calculated. Which condition is used as control.
- Fig. 1J. Indicate which condition is used as a reference.
- Fig. 2A. Font size is extremely small.
- Fig. 3A. indicate in the figure legend where this data comes from. (RNA-seq?)
- Fig. 3F,G,H figure legend does not correspond with the panels shown in the figure
- Panels 5E, F figure legend does not correspond with the panels shown in the figure.

Reviewer's comments

General response: We thank all reviewers for the critical but helpful comments that enlighten us to carry out new experiments to address reviewers' comments and more correctly describe our findings. We have since extensively revised our manuscript. Please find point-to-point response below:

Reviewer #1 (Remarks to the Author):

In the manuscript "Cryptococcal Hsf2 governs mitochondrial ROS homeostasis by regulating respiration independent of UPR", the authors identify a new role for a *C. neoformans* HSF family member that participates in regulation of mitochondrial gene expression in response to oxidative stress. The authors provide compelling evidence that loss of Hsf2 sensitizes *C. neoformans* to temperature and oxidative stress, and is required for mitochondrial-dependent carbon source utilization at elevated temperature. This conclusion is supported by transcriptomic, metabolomics and IP-MS. Interestingly, Hsf2 binds an A-rich consensus sequence, and the DNA binding activity appears to be dependent on the oxidation state of a cysteine in the DNA binding domain, making Hsf2 a sensor of mitochondrial redox balance. The experiments are well-designed and, in general, the data support the conclusion. Below are a few weaknesses that should be addressed.

1. Oxidation of Hsf2 leads to an increase in MW beyond that which would be expected by oxidation, suggesting potential dimerization or aggregation. This data, in light of the fairly low resolution MEME logo for Hsf2, may suggest that DNA binding is multimer dependent. Perhaps analysis of the motif as two motifs (two, 6-base motifs) would yield a stronger consensus? Further, DNA IPs are presented, but EMSAs might help to determine if DNA binding is multimer dependent. If so, the effect of oxidation on multimerization may be warranted. There is data suggesting mammalian HSF1 and HSF3 bind as trimers.

Response: We appreciate the comments. We identified an increase in molecular weights of CnHsf3 (renamed as suggested by reviewers 2 and 3), suggesting the possibility of oxidation and protein oligomerization or aggregation. Previous studies of Hsf1 proteins in other model systems demonstrated that Hsf1 forms oligomers for DNA binding to regulate gene expression (Gomez-Pastor et al., 2017). Recent studies of human Hsfs also indicated the formation of oligomer topology during transcription activation (Gomez-Pastor et al., 2018; Hentze et al., 2016; Sarge et al., 1993). We previously utilized an approach of HSF motif analysis adopted from other species to analyze HSF motifs of CnHsf3, hHsf1, hHsf2, ScHsf1 and CnHsf1 and found an GAA with poly-A-like motif, with a statistically E value of 8.2×10^{-44} .

We have also utilized the two-motif approach that yielded a statistically insignificant result. In the revised manuscript, we performed EMSA as suggested that showed CnHsf3 forms aggregated or oligomerized structures. We also showed that the oligomerization or aggregation is protein concentration or oxidation dependent. This finding validated the likely mechanism of CnHsf3 function. The statistical analysis of the binding motif has been and the description of EMSAs has been added. The results and discussion sections have also been updated accordingly. Please see updated Figures 1, S3C, S9A, and text lines 151-152, 157-158, 344-346, 395-403.

2. There's approximately a 10% increase in mtROS in Hsf2 consistently across the time course presented in Figure 3H and I, but a 40% increase in cytoplasmic ROS. This is interesting, and it would be important to understand if cytoplasm ROS is unchanged as mtROS in control experiments as in Fig 3H. The authors state that the cytoROS accumulation is caused by mtROS accumulation, but the data as presented don't necessarily support this. Also, is 10% increase in mtROS similar to that seen by treatment with an uncoupling agent like antimycin A? It would be nice to see a positive control for mtROS induction and its effect on cytoROS to appreciate where this hsf2 effect is on magnitude scale.

Response: We have performed new experiments with proper controls to quantify cytoROS and mtROS. We previously assessed cell population and signal intensity to estimate ROS production and detected an approximately 10% increase (cell population) using mtROS or cytoROS dyes (Figures 3H, 3I for mtROS and S6A for cytoROS). But higher intensities were detected from cytoROS than mtROS (Figures 3J and 3K), suggesting cells produce more cytoROS than mtROS in response to thermal stress. In the revised manuscript, we performed the flow cytometry analysis of ROS in response to antimycin A treatment. In this result, we showed that antimycin A treatment at normal temperature resulted in approximately 48% and 56.6% increases in wild-type and Cnhsf3 mutant strains, respectively. There was approximately 8% of induction in the Cnhsf3 mutant cells, similar to heat shock treatment. However, the signal intensity of antimycin A treated Cnhsf3 mutant cells had about 50% of induction in mtROS. It is hypothesized that antimycin A causes mitochondrial dysfunction and ROS generation partially and it does not fully recapitulate the response from heat stress. We have updated the figures and text to present our new results accordingly. Please see Figures S6A, S8C, S8D, S8E, and S8F. Please also see text lines 220-221,302-304.

3. The authors conclude that the HSF2C130A mutant is defective in DNA binding, but in light of comment 1, DNA binding should be tested directly either by ChIP, IP or EMSA, or SPR, all of which are shown for the DBD of Hsf2. It would be ideal to determine the contribution of direct anti-oxidant function apart

from transcriptional activation to the hsf2 mutant phenotype. Is there a mutation that would abrogate DNA binding, but that would leave the oxidant-sensitive cysteine intact?

Response: We have since performed both the localization of the CnHsf3^{C130A} protein and CHIP-PCR analyses, since the effect of CnHsf3^{C130A} may act through protein mislocalization and the inability to bind DNA. The new data showed that the mutation of 130 residue has no effect on CnHsf3 nuclear localization. CHIP-PCR demonstrates that the C130A mutant protein cannot bind to the promoter region of its target gene. Together with downregulated CNAG_09000 gene expression in the CnHsf3^{C130A} strain, our data demonstrate that Cys130 plays an essential function in the CnHsf3-DNA interaction. In addition, the NLS and MTS signal mutation strains, which abolished protein trafficking but with the intact 130 cysteine residue, showed high ROS production but do not protect cells from mitochondrial stress. This finding suggests that it is the gene regulation, rather than anti-oxidant effect, of CnHsf3 in modulating mitochondrial ROS homeostasis. We have updated relevant statements in the Discussion. The future research effort will be focused on the generation of DNA binding mutants and their characterization. Please see Figures 5H, 7J, S9B, S9C and S9D. Please also see text lines 286-288, 352-357, 467-472.

4. The authors conclude electron leakage from the etc is the source of ROS. It would be helpful to identify the specific results that suggest this. It is unclear which data support this conclusion.

Response: Our statements were partially based on various previous studies indicating that mitochondrial ETC dysfunction results in electron leakage to oxygen and superoxide anion generation. As the subject has not been the focus of our current studies, we did not address it through experimentation. We have since revised the text in Results and Discussion. Please see lines 221-223, 454-457.

5. The authors talk a lot about MQC, but the direct link between the reported data and how they support a defect in MQC is unclear. This could be made more apparent in the results.

Response: We appreciate the comments. MQC is defined to operate through the coordination of various cellular processes, including proteostasis, biogenesis, and dynamics and mitophagy to ensure cell homeostasis. Our studies demonstrated that CnHsf3 regulates the TCA cycle and biogenesis of ATP. In addition, CnHsf3 plays an important role in maintaining mitochondrial morphology. We have since revised the text to lessen the emphasis on MQC and describe our findings more directly. Please see lines 175-213.

6. The authors use only parametric statistical tests, including for the ordinal data that is scoring mitochondrial morphology. They should consider non-parametric tests for these data.

Response: We have since added non-parametric statistical analysis in mitochondrial morphological characterization and updated the figure and text accordingly. Please see Figure 4I and the legend.

7. The authors include "independent of the UPR" in the title. I think this detracts from the fundamental role of Hsf2 in redox sensing and mt function

Response: We agree and revised the title accordingly.

Reviewer #2 (Remarks to the Author):

In this manuscript, Gao et al. investigated the function of Hsf-like protein 2 (Hsf2) in mitochondrial quality control and homeostasis in *C. neoformans*. They found that expression of HSF1 is highly upregulated in response to high temperature (40°C) and Hsf2 is involved in thermotolerance of *C. neoformans* at 40°C. Interestingly, the ChIP-seq analysis indicate that Hsf2 binds to the promoters of genes involved in metabolism, but not in protein chaperones, suggesting that Hsf2 regulates heat shock response via UPR-independent manner. Subsequently, the authors showed that Hsf2 is required for mitochondrial metabolism and responding to mitochondrial reactive oxygen species (mtROS). Notably, the authors found that Hsf2 is both mitochondrial and nuclear targeting transcription factors and activate transcription of genes involved in electron transport chain to detoxify ROS. They also showed that Hsf2 is a sensor and modulator for mtROS homeostasis under various conditions that lead to mitochondrial dysfunction. Finally, the authors demonstrated that oxidation of cysteine residues in its DNA binding domain reduce Hsf2 binding to the mitochondrial genome.

Overall the authors provided a large amount of molecular genetic and biochemical data for the novel roles of Hsf2 in mitochondrial quality control and homeostasis in *C. neoformans*. The data presented by this study could be very informative and useful for those working in this pathogen. Having said that, I have the following major criticism, which makes me less enthusiastic for this work. The authors did not provide any in vivo data regarding the functions of Hsf2 in the pathogenicity of *C. neoformans*. If Hsf2 is a universal mitochondrial integrity modulator in *C. neoformans*, does inhibition of Hsf2 actually impact the pathogenicity of the pathogen? Previously, the Robin May group demonstrated that *Cryptococcus gattii* adopts a mitochondrial tubularization in response to host ROS and these fungal cells facilitate the rapid growth of neighboring *C. gattii* cells with non-tubular mitochondria, leading to increased pathogenesis of

infected *C. gattii* population (Voelz et al. 2014 NatCommun). In this manuscript, the authors mentioned that deletion of HSF2 resulted in dysregulation of mitochondrial morphology in *C. neoformans*. Therefore, although *C. neoformans* and *C. gattii* are two different *Cryptococcus* species, this is one of the issues that the authors should think about for the potential roles of Hsf2 in the pathogenicity of *C. neoformans*. Collectively, the lack of data showing the role of Hsf2 in the pathogenicity of *C. neoformans* markedly reduces the significance of this work.

Response: We thank the reviewer for the positive response and the critical but helpful comments. In the revised manuscript, we have performed mouse infection experiments that showed the Cnhsf3 Δ strain is significantly attenuated in fungal virulence. Please see Figure S1C. We have also expanded literature citation by including the article by Voelz et al. (Nat Comms, 2014). Please see lines 440-441.

I have the following additional comments and concerns for this work.

1. The authors did not appear to properly cite previously published papers and discuss them, which are relevant for this work. For example, here the authors mentioned that they identified three Hsf1-like proteins and named Hsf1, Hsf2 (CNAG_04036), and Hsf3 (CNAG_04176). However, these three Hsf1-like proteins in *C. neoformans* have been already reported in the previous papers (Jung et al. 2015 Nat Commun; Yang 2017 Genetics). In these previously published papers, the three Hsf proteins were named as follows: Hsf1 (CNAG_07460), Hsf2 (CNAG_04176), and Hsf3 (CNAG_04036). Therefore, as these previous publications exist, the authors should follow their gene names to avoid any potential confusion among readers. In addition, Yang and colleagues previously reported the essentiality of Hsf1 in the growth of *C. neoformans* by conditional promoter replacement and sporulation analysis of heterozygous HSF1/hsf1 mutants (Yang 2017 Genetics). They also showed that expression of HSF1 is downregulated, which is in agreement with the data presented by the current work, and overexpression of HSF1 increases the thermotolerance of *C. neoformans*. Yang et al. also demonstrated that Hsf1 can physically bind to the promoter of SSA1 through ChIP-PCR analysis, which is in agreement with the data presented by the current work.

Response: We apologize for not including these references, which is not intentional, and have since expanded literature citation by including the relevant published studies. We have revised gene names to be consistent with those used in the research field. Additionally, we have repeated GalP-Hsf1 experiments with new plasmids to further distinguish differences in regulation between CnHsf1 and CnHsf3. Please see Figures S1 and lines 125-127.

2. Here the authors tested the thermosensitivity of the hsf2 mutant at 40°C, which is not the host biological temperature. What is the role of Hsf2 in the growth of *C. neoformans* at 37°C?

Response: In the revised manuscript, we tested temperatures at 25°C, 30°C, 37°C, and 40°C (Please see Figure S1). The *Cnhsf3Δ* strain demonstrated abnormal mitochondrial morphology, increased fragmented mitochondrial structures, and decreased in mitochondrial membrane potentials at the lower temperatures, it did not show any growth defects at 25°C, 30°C, and 37°C.

We demonstrated that complementing the *Cnhsf3Δ* strain with *Sod2* and mitochondrial ETC genes could rescue the growth impairment at 40°C. This suggests that it is the mtROS that causes cell growth defects, rather than higher temperatures. 25°C, 30°C, and 37°C are not mitochondrial stress-inducing temperatures for *C. neoformans*. Additionally, at the lower temperatures, mtROS remains at the same level as the wild type strain, thus no growth phenotype difference was detected. We appreciate the fact that the host temperature is at 37°C and the cause of virulence attention in the *Cnhsf3Δ* strain is not directly relevant to the thermal stress, but immune killing. Please see Figures S1F, 3H, 3L, 3M, S6E, S6F, S6G and S6H and lines 216-218, 227-234,

Reviewer #3 (Remarks to the Author):

General comments:

In this manuscript, the authors have identified and characterized CnHsf2, an HSF-like protein or a protein possessing domains related with HSF DNA-binding domain in *Cryptococcus neoformans*. Several vertebrate HSF-like proteins have been identified and characterized, but their roles are not well known. The authors first determined that CnHsf2 regulates expression of nuclear-encoded genes involved in the TCA cycle and mitochondria-encoded electron transfer chain genes, and protects cells from heat shock and mitochondria oxidative stresses. Localization of CnHsf2 in both the nucleus and mitochondria and its roles in regulation of both nuclear-encoded and mitochondria-encoded genes under stress conditions are unusual. These exciting conclusions are derived from a lot of experiments with high quality. There are several points below that should be clarified before publication.

Specific comments:

1) The authors imply that CnHsf2 is an ortholog of human HSF-like protein, named HSF5, based on the phylogenetic analysis (Fig. S1A). This point is interesting and important because this study first show roles of HSF-like proteins in a cell. In the introduction (lines 81-83), the authors define HSFs as transcription factors that bind to the HSE. Indeed, HSF1 to HSF4 bind to the HSE, but the HSE is not bound by HSF-like proteins, such as human HSFX,

HSFY and HSF5, which possess domains related to the DNA binding domain of HSFs. Therefore, an explanation should be required to distinguish the HSF-like proteins from HSFs in the introduction.

Response: We appreciate the reviewer's correction in using the term "HSF-like". The HSF family is defined as a protein family with classical HSF DBDs (Gomez-Pastor et al., 2017). hHSF members included HSF1, HSF3, HSF4, HSF5, HSFY, and HSF3, whereas HSF3 was only found in mice. While HSF1-HSF4 were in-depth investigated, HSF5 was only analyzed at the transcriptome level between wild type and HSF5^{-/-} in zebrafish, without the description of detailed binding motifs. Moreover, the HSFY and Y are located on sex chromosomes, and their functions and binding motifs remain unknown. We here show that the CnHsf3 possesses the classical HSF DBD and is significantly similar to the HSF family. Our early description of CnHsf3 being an HSF-like protein is clearly incorrect. We have since revised the statements, as suggested. Please see lines 136-146.

2) It is important to show detailed information to determine whether CnHsf2 is an ortholog of human HSF5 or not. First, the authors should show an alignment of the full-length amino acid sequences of human HSF5 and CnHsf2 in Fig. S1. Second, they should examine *in vitro* binding of CnHsf2 to a consensus heat shock response element (HSE) and the adenine-rich sequences shown by ChIP-seq data in Fig. 1E by performing EMSA. It is also important to show whether CnHsf2 binding to these sequences and its oligomerization is induced during heat shock. Third, it would be better to determine whether human HSF5 is able to substitute CnHsf2 for target gene expression or some phenotypes.

Response: We previously performed DBD similarity and phylogenetic analysis to demonstrate that CnHsf3 (previously CnHsf2) is an HSF family member protein, and the phylogenetic relationship tree demonstrated the closest relative of HSF to CnHsf3 is hHsf5. We have included the BLAST alignment in the new Figure S2C. We have also performed EMSA analysis as suggested. This data demonstrated that CnHsf3 can bind to its target promoter sequence. The competition control group using an unlabeled probe (cold probe) could compete with the target promoter, whereas typical HSF target promoters (HSE containing Hsp70 and Hsp90 promoters) were unable to compete for the binding. The data also demonstrated that CnHsf2 did form aggregation or oligomerized structure, as detected oligo-bound protein signal showed multiple protein-DNA complex species, rather than a single band. We also showed that the oligomerized or aggregation is protein concentration or oxidation dependent. Moreover, we constructed a hHsf5 cDNA expressing strain in the Cnhsf3 knockout strain and showed that the expression of hHsf5 complements the heat-sensitive phenotype of the Cnhsf3 knockout strain. Please see revised Figures

S2C, S2D, S3D and S9A and lines 136-139, 158-161, 344-346.

3) The authors claim that “Hsf2 has the GAA-like motif enrichment in adenine”. This statement may mislead readers about its binding sequences. It would be better to mention that the sequences are adenine-rich, but is different from repeats of nGAAn motif.

Response: We agree and have modified the manuscript accordingly. Please see lines 148-152.

4) The authors claim that CnHsf2 localizes in the mitochondria and nucleus in Figs. 4 and 5. However, it seems by western blot data in Fig. 4C that only a little fraction of CnHsf2 localizes in the mitochondria. They should compare CnHsf2 levels in the nucleus, cytoplasm, and mitochondria in the same blot, and should determine whether CnHsf2 moves to the nucleus and mitochondria during heat shock or not. In addition, immunofluorescence data in Fig. 5D clearly showed exclusion of CnHsf2-NLS_mut-GFP. They should also show localization of CnHsf2-MTS_mut-GFP. If this construct is too large to translocate to the mitochondria, they can stain overexpressed CnHsf2-MTS-FLAG and CnHsf2-MTS_mut-FLAG in cells.

Response: We appreciate the comments. Cell fractionation to isolate mitochondria, cytosol, and nuclei remains technically challenging in studies of *C. neoformans* due to its polysaccharide capsule structures. In the revised manuscript, we employed a mitochondrial protein isolation protocol published by others (Chang et al., 2019; Horianopoulos et al., 2020). We quantified the protein ratio between CnHsf3 and Tim44 that shows an insignificant alteration at 30°C and 40°C, suggesting similar CnHsf3 levels of in mitochondria. In addition, we constructed CnHsf3-GFP::Tim44-mCherry, CnHsf3mut-GFP::Tim44-mCherry, and CnHsf3C130A-GFP::Tim44-mCherry strains and performed fluorescent microscopy analysis. CnHsf3 is predominantly localized in the nuclei, but it was also detected in mitochondria, colocalizing with Tim44. Taking account of all the data from microscopic examination, protein co-IP, protein co-IP/MS, ChIP-seq, ChIP-PCR, and mitochondrial immunoblots, we have conclusively demonstrated that CnHsf3 is a mitochondrially localized transcription factor. Please see Figures 4C-F, 5B and C, S7A and C, and S9C, and text lines 242-257, 276-277, 352-357.

5) The CnHsf2 mRNA level was induced during heat shock (Fig. 1A). However, it is unclear what “activation of CnHsf2” means (P10, line 283; P11, line 316). Is CnHsf2 protein level upregulated during heat shock? Is DNA-binding activity (to nuclear-encoded genes) elevated during heat shock? Does oligomeric form of CnHsf2 change during heat shock? The authors should mention these points by showing experimental data.

Response: We thank the reviewer for the suggestion. We agree that the description “activation of Hsf2” misleads readers. We have provided compelling evidence demonstrating that CnHSF3 gene and protein expression is induced in response to heat treatment, with increased bindings of CnHsf3 to mitochondrial gene promoters. Therefore, we have revised our manuscript writing to be more explicit and clear about this conclusion. Please see Figures 1A, S1A, S3A, S3B, S9A and lines 116-118, 156-157, 344-346, 401-403.

6) In Fig. 1F, did the authors analyze unstressed cells or heat-shocked cells? Explain this point in the legend.

Response: The ChIP-seq analyses were performed using heat-shocked cells. We have since revised the figure legend and Material and Methods accordingly. Please see text lines 526-528, 760-761.

7) It would be better to re-consider the name “Hsf2”. This is not related with HSF2 in vertebrates, but could be “CnHsf5”. CnHsf1 is an ortholog of vertebrate HSF1, but CnHsf2 and CnHsf3 (in this manuscript) are not orthologs of vertebrate HSF2 and HSF3. Readers in the wide fields (for nature communications) may be confused with these names One possibility is that Hsf2 and Hsf3 could be HSF-like protein 1 (Hsf1) and HSF-like protein 2 (Hsf2).

Response: We thank the reviewer for the suggestion and renamed cryptococcal HSF genes to be consistent with previously published ones: CnHsf1 (CNAG_07460), CnHsf2 (CNAG_04176), and CnHsf3 (CNAG_04036). Please see lines 113-116.

Reviewer #4 (Remarks to the Author):

In the manuscript by Gao et al., the authors characterized a new HSF protein (Hsf2) in the pathogen *C. neoformans* that is homolog to human HSF. They conduct experiments in the presence of Heat Shock (HS) or various oxidants and demonstrated that cnHsf2 reacts to the accumulation of ROS which activates the localization of cnHsf2 into the nucleus and mitochondria where it regulates the expression of genes related to the TCA and ETC. Deletion of Hsf2 is viable but causes growth defects in the presence of stress which the authors attributed to a defect in the regulation of mitochondrial quality.

While this is an interesting study that provided new insights into the Heat Shock Factors field, the study is very limited to the characterization of cnHsf2. The most important concern is that the study lacks significance in understanding how this mutation, if relevant, influences pathogenesis by this fungus. In other words, is this protein relevant in any way for the pathophysiology of *C.n*?

Without biological significance, the scope of the study is very limited.

Response: We thank the Reviewer for the generally positive response towards our studies. For the specific question raised here, we have carried out new animal testing and determined that CnHsf3 is indeed involved in fungal pathogenicity in mice and this role is independent of its function in regulating thermal stress. The animal study underscores the importance of CnHsf3 and justifies our study efforts. Please see Figure S1C and text lines 120-121.

Major concerns:

As of general comments, several experiments lack important controls, in several cases samples are compared inappropriately and conclusions are not always supported by their results. The authors need to (a) provide individual data points for all graphs and indicate the statistical analyses performed in the figure legend of each figure. (b) All spot assays must be accompanied by growth curves where replicas and statistical analyses can be performed to better quantitate the impact of genetic mutations and/or treatments.

Response: We thank the Reviewer for the critical comments and have repeated relevant experiments to include proper controls and individual data points for all graphs, with necessary statistical analysis. We have also repeated spotting assays. Please see Figures 1 to 7 and Supplemental Figure 1 to 11.

In Fig. 1B, the authors stated “the loss of HSF2, but not HSF3, attenuates growth at 40°C (Figure 1C), suggesting Hsf2 and Hsf3 have distinct roles as HSR. The effect on growth attenuation between Hsf2D and Hsf3D is very modest. Since the phenotype of Hsf2 seems to be critical for concluding that Hsf2 participates in HSR, the authors should conduct an experiment where such phenotype can be quantitated. A growth curve comparing 30C vs 40C should be shown to indicate the extent of the attenuation and the differences between Hsf2 and Hsf3. Please include a Hsf1 deficient mutant (in the case that Hsf1D is not viable) in the analysis as a reference (positive control), since the comparison with the ‘main HSF’ is necessary to draw conclusions about how important Hsf2 is in the HSR.

Response: We have performed a growth comparison in liquid culture at 30°C and 40°C. In consistence with the spotting assays, the Cnhfs3 deletion strain (previously Cnhfs2) exhibited a significant growth defect at 40°C. Please see Figure S1B.

We have also generated a Cnhfs1 mutant strain to verify its phenotype. We replaced the endogenous promoter of CnHSF1 with an galactose inducible promoter that allows examination of CnHSF1 should it be an essential gene.

The GalP-CnHSF1 strain can only grow in the presence of galactose, which rules out its use as a positive control. Please see Figures S1E, S1D and S1F. Please see lines 125-128.

Regarding the structural comparisons shown in Fig.1, The comparisons presented seem random, with no description of whether there is any similarity with C.n Hsf1. It is unclear how the authors hypothesize that the mechanism of Hsf2 response might be similar to that of other HSFs. The authors should comment on (1) the structural homology between C.n Hsf1 and Hsf2, (2) whether this comparison resembles the differences between mammalian Hsf1 and Hsf2, (3) How different/similar C.n Hsf2 is compared to other HSFs in yeast and/or mammals.

Response: We appreciate the comment. The CnHsf3 structure was modeled and predicted based on sequence similarity and residue properties to other HSF proteins using the algorithm from <https://swissmodel.expasy.org>. In the revised manuscript, we have included CnHsf1 and CnHsf2 in detailed alignment and statistical analysis. CnHsf1 shares 51.85% and 50.00% identities with hHsf1 and hHsf2, respectively, whereas CnHsf3 shares 32.41% and 26.42% identities with hHsf1 or hHsf2. This complements the initial bioinformatic analysis demonstrating that CnHsf3 shares relatively low similarities to human Hsf1 and Hsf2. However, the DNA-interacting helix showed a strong sequence conservation. We have provided a more clear discussion of the differences in the revised manuscript. Please see Figure S2 and text lines 139-146.

The authors conclude that Hsf2 regulates the HSR through mechanisms different from the UPR. Please include a comparison with Hsf1 as a positive control to compare the response of Hsf2D with the HSR expected from a canonical HSF. Please include a quantitative representation of example genes for Hsf2 as shown in Fig. 1I for Hsf1.

Response: We previously performed ChIP-seq analysis to show that the regulation of CnHsf3 is different from CnHsf1, where CnHsf1 binds to the promoter regions of protein chaperones and CnHsf3 does not. We also tested the expression of four protein chaperones in the Cnhsf3 mutant strain, validating that CnHsf3 does not regulate protein chaperones at the transcription level. In the revised manuscript, we further tested the binding of CnHSFs to promoters of two metabolic genes MDK1 and HXK1. While CnHsf3 showed strong enrichments at these two promoters, CnHsf1 remained low in promoter binding. We further constructed two overexpressing strains, TEFP-HSF1 and TEFP-HSF3. Induction in the CnHSF1 gene expression caused robust

activation of SSA1 and HSP90 gene expression, but not by CnHSF3. We also performed EMSA assays that demonstrate CnHsf3 binding to its target promoter sequence. Data from ChIP-seq, ChIP-PCR, knockout of CnHSF3, qRT-PCR and overexpression experiments, and EMSAs all indicate that CnHsf3 is an atypical HSF. Please see Figures 1F to 1I, and S3A to S3D. Please see lines 153-161.

Fig. 2A Please explain how the KEGGs are nearly identical between wt and Hsf2D cells, especially for those genes related to carbon metabolism. Shouldn't the authors expect differences in the pathways that are supposed to be Hsf2 dependent?

Response: The heat shock environment at 40°C is an extremely harsh and lethal condition for cells, with alterations in many intracellular pathways including CnHsf1, CnHsf3, translation, transcription, fatty acid biosynthesis and others. The CnHsf3 regulation axis only represents a small portion of the global cell responses to heat. In the study, our KEGG analysis showed global transcriptome alterations. The CnHsf3 may not fully represent whole cell responses to heat. In addition, Figure 2 and the following data indicated that CnHsf3 modulates gene expression of TCA cycles and ETC, downstream of glycolysis. These special circumstances may explain why a similar KEGG profile was found between the wild type and the Cnhsf3 mutant strain.

The authors need to better explain why there is no growth phenotype in Hsf2D cells grown in YNB. They indicated that 'Cells grown in the absence of six-carbon sugars were partially rescued at 40C'. but it seems that there is no difference in growth at all and that any growth defects are completely abolished in the absence of sugar. Growth curves should be shown to better determine the extent of the attenuation or amelioration in growth effects by carbon source and should be compared with a Hsf1 deficient mutant.

Response: In the revised manuscript, we have performed the liquid growth assay to quantify alterations in cell growth. We also updated the Material and Methods to include the description of the YNB medium. YNB is composed of a yeast nitrogen base, without sugar and other supplements. In this medium, fungal cells undergo nitrogen metabolism instead of carbon metabolism, bypassing TCA cycle-ETC axis. Given the function of CnHsf3 in regulation of the TCA cycle and ETC, the utilization of YNB was used to confirm the results from transcriptome and metabolome. We cannot compare with a CnHsf1 deficient mutant, as CnHsf1 is an essential gene. The manuscript has been modified to clarify the comment. Please see Figure S4C. Please see text lines 180-186 and supplemental file lines 22-24.

In fig. 3C the authors indicated "we employed random hexamer qRT-PCRs to

measure mitochondrial encoding ETC gene expression.” To justify that they did not find any mitochondrial gene in their RNA-seq studies. However, to the knowledge of this reviewer, there are no reports in the literature indicating that oligo(dT) for reverse transcription could result in the insufficient synthesis of mitochondrial cDNA. The authors claim they are analyzing mitochondrial encoding ETC gene expression just by using hexamers to synthesize cDNA from whole cell extracts. However, it is unclear how the authors distinguish between nuclear encoded and mitochondrial encoded genes just by changing from oligodT to hexamer amplification. Nowhere in the methods section it is described such distinction. Unless the authors purify mitochondria and isolate RNA from mitochondria it would be extremely difficult to distinguish between the RNAs that are coming from nucleus or mitochondria when preparing cDNA from whole cell extracts using either oligodT or hexamers.

Response: In the article(Butow et al., 1989; Schafer et al., 2005) Mihara, et al., Mol Cell 2003, the authors found that yeast mitochondrial mRNAs are not polyadenylated and no PAP activity was identified in the mitochondria. Instead, mitochondrial mRNAs carry a conserved dodecamer sequence, AAUAA(U/C)AUUCUU, at their 3'-ends. Similarly, mitochondrial mRNAs are also not polyadenylated in fission yeast and they have a C-rich motif near the 3'-end. Therefore, polyadenylation of mitochondrial mRNAs may be restricted to the higher eukaryotes. In the revised manuscript, we compared gene expression of the mitochondrial gene using oligo(dT) and random hexamers. Please see Figure S5C. In our hands, cDNAs of mitochondrial genes remained low and undetected using oligo(dT), but robust gene expression was detected using random hexamers. The manuscript was revised by including proper references. Please see lines 191-195

The conclusion that 'six of seven tested ETC genes are positively regulated by Hsf2' is not accurate. The authors only compared wt vs Hsf2D at 40C and what they observed is that those 6 genes are partially repressed in Hsf2D compared to wt. However, some of those genes seem to also be repressed in WT at 40C compared to 30C or not changed indicating that the response observed in those genes does not correspond to a positive regulation by HSF2. The authors need to statistically compare all conditions to better determine which genes respond to Heat shock (HS) and whether they are regulated by Hsf2.

Response: We thank the comments and accept that the description might mislead readers in the previous manuscript. We have modified the statement. Mitochondrial gene expression is sophisticated with many unknown transcription regulation factors and mechanisms. We agree with the reviewer that the gene expression of mitochondrial encoded genes responds to temperature shifts, with repression, induction or constant expression. While we agree that comparing all conditions to better determine genes responding to

global heat shock response would be more informative, it is clearer and more straightforward for readers to understand CnHsf3 function at its current format. Nevertheless, we have modified our relevant statements. Please see lines 195-197.

Fig. 3D and Fig. 6E. Data representation is confusing. Using IgG negative controls is a standard in ChIP experiments. The authors should include IgG negative controls and data should be relativized to IgG. Then the authors can normalize their data based on % input but a negative IgG control should be used. There is no way to determine if ChIP signals are specific to the antibody without a negative control. Gene CNAG_09004 is not tested in Fig. 3C. Please provide an explanation why this gene was chosen.

Response: We have since repeated all ChIP-PCR using IgG as controls. Please see Figures 3D, 6D, 6E, 6I, 7A, S3A, S3B, S8K, S8L, S9D. In consistence with the previous experiments, we detected significant enrichment at the promoter regions of tested mitochondrial genes. While we cannot test all mitochondrial gene expression using ChIP-PCRs and RT-PCRs, genes in Figure 3 were chosen unbiasedly to illustrate the binding and regulation properties of CnHsf3. Additionally, CNAG_09004 encodes a subunit in complex IV, which is irrelevant to the following studies, it was unbiasedly processed for representative purposes. Please see Figures for IgG controls, and lines 773-776 in Material and Methods.

Fig. 3E. Hsf2D+QCR9 should be in the same plate as Hsf2D+QCR9+NDUFA5 to compare on the same plate the benefit of co-expressing with NDUFA5. Since spots are not on the same plate is it impossible to compare whether co-expressing QCR9 with NDUFA or CNAG_09000 have similar or different effects. The authors need to provide growth curves to better quantitate (including statistics) the impact on growth of the different ETC expressing genes. In addition, since co-expression with only nuclear encoded genes (based on the authors conclusions) seems to restore growth deficits, then the impact on mitochondria ETC by Hsf2 does not seem to play a key role in the temperature sensitive phenotype.

Response: In the revised manuscript, we included all growth assays in liquid media. In previous and current manuscripts, we have shown that expression of the nuclear-encoded gene QCR9 (subunit of complex III) did not rescue the growth phenotype under high temperatures, but only rescued the phenotype when grown in the presence of antimycin A, which predominantly targets complex III. To rescue the growth in heat-shocked condition, both subunits from complex I and III are required. Additionally, mutating MTS on CnHsf3 significantly impaired cell growth during heat shock, demonstrating its mitochondrial function is also critical in temperature stress response. Please see

Figures 3E and S5I to S5K.

Fig. 3M. Without growth curves and cultures with and without NAC simultaneously compared is difficult to assess whether there is a difference in growth in Hsf2D cells at 40C treated with NAC or not.

Response: Cell growth assays are now included. Please see Figure S6F.

The authors claim that ‘Hsf2 responds to ROS overload of the mitochondrial matrix rather than an elevated temperature.’ The authors do not provide data to support this conclusion. They only showed alterations in ETC gene expression and ROS when comparing Hsf2D at 40C. To claim that Hsf2 does not respond to elevated temperature (independent of ROS) the authors should show that no changes in HSF2 genome wide binding patterns occurs when conducting HSF2-ChIP-experiments in the presence of HS + antioxidants

Response: The CnHsf3-ChIP-experiments in the presence of HS + antioxidants were performed. Please see Figure 7A. Sod2 is a well-defined and -characterized antioxidant that detoxifies mtROS in the mitochondrial matrix (Reference: Sod2 in Mitochondrial Dysfunction and Neurodegeneration)(Flynn and Melov, 2013). We have revised our statements and added the citation to support our conclusion. Please see lines 224-235.

Fig. 4C. The authors need to load 30C and 40C samples in the same gel to compare whether HS alters the amount of Hsf2 found in the mitoc. This seems to be important to understand the mechanism by which Hsf2 regulates mitoc stability under HS. The authors did not show anywhere in the manuscript that HS or any other stress increases the localization of Hsf2 in the mitoc (as it would be expected).

Response: We agree with the reviewer and have performed the immunoblot on the same gel. We also used the Tim44-HA as a control, and compared the relative protein ratio of CnHsf3/Tim44-HA between two temperatures. In the data, we did not detect any significant changes in CnHsf3 protein levels when shifting from 30°C to 40°C. Please see Figures 4C and S7A.

Fig. 4E. This experiment should have been carried out in the presence of wt HSF2-FLAG and loaded in the same gel to compare side by side the impact of mutating the MTS. Without this important control the reviewer cannot see whether this mutation prevents Hsf2 from getting in the mitoc.

Response: We have now provided the same gel analysis. Please see Figure 4E. While we detected the CnHsf2-FLAG from the control sample, the CnHsf3

MTS^{mut} cannot be detected from the mitochondrial isolation.

Fig. 4F. should also include wt Hsf2-Flag as a positive control. Not showing a band in the HA blot does not necessarily mean there is no interaction since absence/presence of signal is relative to the time of exposure. Without a reference to demonstrate that the relative difference with a positive control is difficult to assess whether the MTS mutation does impact or not binding.

Response: We have now provided the proper control group. Please see Figure 4E. We showed that the wild type CnHsf3 could interact with Tim44-HA, whereas MTS mutation protein failed to do so.

Fig. 5B. Hsf2-NLSmut should be included in this experiment to demonstrate mitochondrial localization and to be compared to Hsf2-MTSMut within the same gel. Same applies to Fig. 5C. In addition, samples from 30C and 40C in Fig. 5B should be included in the same gel and imaged simultaneously to determine whether there are temperature-dependent differences in the localization.

Response: We have now provided the proper control group. Please see Figure 5B and 5C. The NLS mutant was compared with the MTS mutant. We detected mitochondrial localization for NLS mutation but not for MTS mutation.

Fig. 5D, Similarly to the previous panels, the authors should have included Hsf2-MTSMut as a control to show nuclear localization and to be compared with Hsf2-NLSmut.

Response: We have now provided the proper control group. Please see Figure S7C. We showed that CnHsf3 is localized in nuclei and mitochondria, whereas the MTS mutation was only detected in nuclei.

The authors indicated that “binding of Hsf2 to mitochondrial DNA is proportional to mtROS production”. Please provide quantitative data to support this conclusion. No experiments or analyses addressing the proportionality of this effect is shown.

Response: Figure 6 demonstrates that the addition of antimycin A causes induction of mtROS production, which is proportional to the incubation time with antimycin A. This suggests that longer incubation of antimycin A enhances the production of mtROS. Having said that, the longer incubation of antimycin A results in higher binding of CnHsf3 to target promoters. The same applies for menadione. Therefore, two well-defined mtROS inducers (antimycin A and menadione) yield similar data, suggesting that CnHsf3 binding to targets depends on mtROS levels. The data are provided in Figure 6. Additionally, the

incubation duration 5 min, 10 min and 15 min at 40°C also enhance the mtROS production and the enrichment at the CNAG_09000 promoter in the meantime. The data are provided in Figure 3I, 3J and S8L. We have revised the statement accordingly. Please see text lines 330-331.

In Figure 7, the authors conclude that oxidation of Cys130 is necessary for the mitochondrial protection by Hsf2 since a Hsf2C130A mutant behaves similarly to hsf2D. The authors need to show that this mutation does only alter DNA binding and not protein localization. It is possible that the Cys130 is a critical residue within the Hsf2 structure to mediate compartment localization and therefore its mutation will generate a protein that is not functional.

Response: We have now provided proper control groups. Please see Figures S9B and S9C. We showed that Cys130 mutation does not affect protein localization and DNA binding was reduced. Please see Figure S9D. A non-CnHsf3-target was used as the control, in addition to IgG as a control.

Minor concerns:

The authors could consider referencing to Hsf2 throughout the manuscript as cnHsf2 since the nomenclature can be confused with mammalian Hsf2.

Response: We have done so in the revised manuscript.

The description of data presented in Fig. 1B is not accurate: the authors indicated “expression of both HSF1 and HSF2 is responsive to elevated temperatures (Figure 1B)”, The expression of Hsf3 is also ‘responsive to elevated temperature. Indeed, Hsf2 and Hsf3 have a similar response to 40C while HSF1 presented an opposite response. The authors should better discuss these differences among the three HSFs.

Response: We have revised the manuscript accordingly. Please see lines 116-128.

Define the composition of YPD and the difference with YNB in the methods section, this is relevant to understand the discrepancy in the grown effects between these media and the experiments where different carbon sources are added.

Response: The information on media has been included. Please see supplemental Material and Methods.

The representation in Fig.2D is difficult to follow. Please provide a

representative result where comparisons can be made quantitatively.

Response: Figure 2D has been updated with a simple version.

In Fig. 5E, Is this experiment conducted at 30C? Provide info in the figure legend. The authors showed a significant reduction in the expression of CNAG_09000 between wt and Hsf2D of ~50%. However, this contradicts the data presented in Fig. 3C where the two graph bars (no statistical data provided) seem to be identical.

Response: The experiments were performed at 40°C. The figure legend has been revised. Figure 3C and 5E consistently demonstrated reductions of CNAG_09000 gene expression between the wild type and the Cnhf3 mutant strain. And we have shown repeatedly that CnHsf3 regulates CNAG_09000 under mitochondrial stress. Please see lines 664-666.

In Fig. 5H the authors indicated “mtROS-producing cell counts and signal intensities of these mutants were significantly greater than those of the wild-type cells (Figures 5G and 5H). This sentence is not accurate. The mutants present higher mtROS compared to Hsf2D cells complemented with HSF2. The authors should consider including statistical comparisons between the mutants and Hsf2D cells too. When doing such comparison, it can be seen that MTSmut produced similar ROS than Hsf2D but NLSmut is worst, which would suggest that the localization in the nucleus plays a more important role than the localization in the mitoc.

Response: We appreciate the comments. We have since included the statistical analysis. In the revised manuscript, we showed a significantly induction of mtROS in the NLS^{mut} strain than in the Cnhf3 Δ strain. Therefore, as suggested by the Reviewer, the nuclear function of CnHsf3 might play a more important role than that of mitochondria localized CnHsf3. We agree that there are additional important mechanistic questions that remain to be addressed, which are the focus of the current research effort in our laboratory. We have revised the manuscript. Please see lines 422-426.

Concentrations for menadione/antimycin ‘ μ M’ should be μ M.

Response: We have made appropriate corrections throughout the manuscript, when necessary.

Fig.6G. showed spots cut out from the plate. Please provide all spots on the same plate so they can be compared with each other.

Response: We have updated the image in Figure 6G.

It would be important to comment whether there is a MTS in other yeast and/or mammalian HSFs? Is this unique to *C.n.* Hsf2?

Response: In response to the comment, we have further analyzed hHsf1, hHsf2, hHSF5, ScHsf1 and CnHsf1. Interestingly, the MTS motif was identified in CnHsf3 and hHsf5, but not in other HSF proteins. We have updated the manuscript statements accordingly. Please see lines 417-419.

Considering the alterations observed in the mitoc morphology in Hsf2D cells even at 30C, it would be expected to see alterations in gene expression of mitoc related genes. However, as shown in Fig. 3C, mitoc ETC genes does not seem to show differences between WT and Hsf2D cells at 30C. The authors should discuss this discrepancy.

Response: We thank the reviewer for the comments. The morphological alternations were observed at 30°C, whereas ETC gene expression remained unchanged. We speculated that this could be due to the lack of a mitochondrial stress environment. Mitochondrial stress at 30°C remains low in comparison with antimycin A and menadione treatment at 30°C. However, when applying stress such as mitochondrial inhibitors, the expression alterations were detected. We have discussed this point in the discussion section. Please see 449-454.

The authors indicated “Menadione triggers enrichment of Hsf2 in the mitochondrial genome, and bindings at CNAG_09000 and CNAG_09009 were ameliorated in a time-dependent manner”. However, Fig. 6I showed a time-dependent increase in Hsf2 binding on those genes.

Response: We showed that antimycin A or menadione could cause the binding of CnHsf3 to mitochondrial genes. The enrichments of CnHsf3 on these gene promoters increase with prolonged incubation time. In our study, exposure to heat, antimycin A or menadione induces the binding of CnHsf3 to mitochondrial gene promoters in a time-dependent pattern. The manuscript has been modified to clarify this point. Please see lines 300-301, 317-319.

Fig. 1A, and 1F provide a numeric range for the color scale

Response: We appreciate the comment. We have modified the manuscript accordingly.

Fig. 1B, add the ACT1 normalization control in the figure legend.

Response: We appreciate the comment. We have modified the manuscript accordingly. Please see lines 757-758.

Fig. 1H. in the legend the authors indicated “ChIP-seq data of the protein chaperones MDH1 and HXK1 are shown”. These genes are not chaperones.

Response: We appreciate the comment. We have modified the manuscript accordingly. Please see lines 531-532.

Fig. 1I. Provide information about how enrichment is calculated. Which condition is used as control.

Response: We have revised the manuscript accordingly. Please see lines 533-536.

Fig. 1J. Indicate which condition is used as a reference.

Response: We appreciate the comment. We have modified the manuscript accordingly. Please see lines 537-539.

Fig. 2A. Font size is extremely small.

Fig. 3A. indicate in the figure legend where this data comes from. (RNA-seq?)

Fig. 3F,G,H figure legend does not correspond with the panels shown in the figure

Response: We have updated the manuscript accordingly. Please see lines 573-574, 586-593.

Panels 5E, F figure legend does not correspond with the panels shown in the figure.

Response: We have revised the manuscript accordingly. Please see lines 664-668.

Thank you all!

Reference:

Butow, R.A., Zhu, H., Perlman, P., and Conrad-Webb, H. (1989). The role of a conserved dodecamer sequence in yeast mitochondrial gene expression. *Genome* 31, 757-760.

Chang, A.L., Kang, Y., and Doering, T.L. (2019). Cdk8 and Ssn801 Regulate Oxidative Stress Resistance and Virulence in *Cryptococcus neoformans*. *mBio* 10.

Flynn, J.M., and Melov, S. (2013). SOD2 in mitochondrial dysfunction and neurodegeneration. *Free Radic Biol Med* 62, 4-12.

Gomez-Pastor, R., Burchfiel, E.T., and Thiele, D.J. (2017). Regulation of heat shock transcription factors and their roles in physiology and disease. *Nat Rev Mol Cell Biol*.

Gomez-Pastor, R., Burchfiel, E.T., and Thiele, D.J. (2018). Regulation of heat shock transcription factors and their roles in physiology and disease. *Nat Rev Mol Cell Biol* 19, 4-19.

Hentze, N., Le Breton, L., Wiesner, J., Kempf, G., and Mayer, M.P. (2016). Molecular mechanism of thermosensory function of human heat shock transcription factor Hsf1. *Elife* 5.

Horianopoulos, L.C., Hu, G., Caza, M., Schmitt, K., Overby, P., Johnson, J.D., Valerius, O., Braus, G.H., and Kronstad, J.W. (2020). The Novel J-Domain Protein Mrj1 Is Required for Mitochondrial Respiration and Virulence in *Cryptococcus neoformans*. *mBio* *11*.

Sarge, K.D., Murphy, S.P., and Morimoto, R.I. (1993). Activation of heat shock gene transcription by heat shock factor 1 involves oligomerization, acquisition of DNA-binding activity, and nuclear localization and can occur in the absence of stress. *Mol Cell Biol* *13*, 1392-1407.

Schafer, B., Hansen, M., and Lang, B.F. (2005). Transcription and RNA-processing in fission yeast mitochondria. *RNA* *11*, 785-795.

REVIEWER COMMENTS

Reviewer #1 (Remarks to the Author):

Overall, I'm satisfied with the responses to my comments and the addition of important data that further support the conclusion.

There remain a few points that should be clarified:

Line 116-117, measured gene expression, not protein expression.

Line 122, animal model doesn't discriminate adaptation defects from interactions with host immunity.

Line 203, 228 use of the word plasmid suggests episomal carriage of the transformed DNA. The methods suggest that these are integrated. Please clarify.

Reviewer #2 (Remarks to the Author):

In this revised manuscript, the authors attempted to address this reviewer's major and minor concerns with additional experiments.

This reviewer previously pointed out a lack of in-host data showing the role of Hsf2 (now renamed as Hsf3 during revision) in the pathogenicity of *Cryptococcus neoformans*, which seemed to be a major concern of reviewer #4 too. Although I appreciate their efforts of performing animal study to address the concern, its experimental method and data are not relevant and disappointing.

First of all, based on their experimental design, the authors did not appear to seriously investigate the role of CnHsf3 in the pathogenicity of *C. neoformans*. For the conventional murine-based virulence assay, one must include a complemented strain (hsf3::HSF3 strain), along with wild-type and mutant strains (at least include two independent mutant strains in the high-throughput assay). But they just tested wild-type and a single hsf3 mutant strain. Therefore, such a minor virulence difference (just 3-4 day survival day difference) between wild-type and hsf3 mutant strains is not convincing enough, although the authors claimed it statistically significant. In particular, the authors used outbred BALB/c mice, instead of conventionally used inbred A/J mice. Furthermore, they did not even perform fungal burden assay with different infected organs and histopathological analysis. Taken together, considering such a minor role of Hsf3 in the virulence of *C. neoformans*, this reviewer is not convinced that Hsf3 plays a significant role in the pathogenicity of *C. neoformans*.

Reviewer #3 (Remarks to the Author):

None.

Reviewer #4 (Remarks to the Author):

The authors were extremely responsive to previous reviewer's comments and the manuscript has significantly improved. The authors included several new experiments, including pathogenicity studies in mice, repeated a large number of critical experiments to validate their conclusions and increased the rigor of the experiments by including proper controls and providing appropriate statistical analyses.

This reviewer is highly satisfied with the new revised version of the manuscript and congratulate the authors for the enormous effort put towards characterizing CnHsf3.

Only two minor comments:

Figure S2 A and C are extremely small and they can't be read.

As part of the response to the reviewers, the authors indicated that antimycin A treated Cnhsf3 mutant cells had about 50% of induction in mtROS and they hypothesized that antimycin A does not fully recapitulate the response from heat stress.” However, text in the revised manuscript lines 302-304 state that “antimycin A treatment fully recapitulated the phenotypes of cells under high temperatures”. Please clarify these contradicting statements.

Reviewer #1 (Remarks to the Author):

Overall, I'm satisfied with the responses to my comments and the addition of important data that further support the conclusion.

Response: We are grateful for the very positive comments.

There remain a few points that should be clarified:

Line 116-117, measured gene expression, not protein expression.

Response: The mistake has been corrected. Please see lines 116-117.

Line 122, animal model doesn't discriminate adaptation defects from interactions with host immunity.

Response: We thank the reviewer for the insightful comment that we fully agree. In the revised manuscript, we have performed additional infection analysis to decipher *CnHsf3* as a pathogenicity regulator. Our new data (with two independent mutant strains and the complemented mutant strain) showed that mice infected with *Cnhfs3Δ* strains exhibited moderate but consistent prolonged animal survival rate and reduced fungal burdens in lung tissue, suggesting attenuated virulence and less pulmonary fitness. We have revised the relevant statements. Please see lines 119-126 and 448-455.

Line 203, 228 use of the word plasmid suggests episomal carriage of the transformed DNA. The methods suggest that these are integrated. Please clarify.

Response: We thank the reviewer for the careful observation. We used a biolistic apparatus during transformation that often results in integrative DNA transformation as demonstrated in the field. We have modified the relevant statements. Please see lines 207 and 234.

Reviewer #2 (Remarks to the Author):

In this revised manuscript, the authors attempted to address this reviewer's major and minor concerns with additional experiments.

This reviewer previously pointed out a lack of in-host data showing the role of Hsf2 (now renamed as Hsf3 during revision) in the pathogenicity of *Cryptococcus neoformans*, which seemed to be a major concern of reviewer #4 too. Although I appreciate their efforts of performing animal study to address the concern, its experimental method and data are not relevant and disappointing.

First of all, based on their experimental design, the authors did not appear to seriously investigate the role of CnHsf3 in the pathogenicity of *C. neoformans*. For the conventional murine-based virulence assay, one must include a complemented strain (*hsf3::HSF3* strain), along with wild-type and mutant strains (at least include two independent mutant strains in the high-throughput assay). But they just tested wild-type and a single *hsf3* mutant strain. Therefore, such a minor virulence difference (just 3-4 day survival day difference) between wild-type and *hsf3* mutant strains is not convincing enough, although the authors claimed it statistically significant. In particular, the authors used outbred BALB/c mice, instead of conventionally used inbred A/J mice. Furthermore, they did not even perform fungal burden assay with different infected organs and histopathological analysis. Taken together, considering such a minor role of Hsf3 in the virulence of *C. neoformans*, this reviewer is not convinced that Hsf3 plays a significant role in the pathogenicity of *C. neoformans*.

Responses: We thank the reviewer for asking us to emphasize the potential impact of CnHsf3 in fungal pathogenicity. We have since carried out several additional experiments to address the reviewer's concern. In addition to the complemented mutant strain, we have included two independent *Cnhsf3* Δ mutant strains in the new round of virulence testing, and the result is consistent in that CnHsf3 plays a role in fungal virulence (supplemental S1). We have also assessed fungal burdens by estimating colony forming units (CFUs) and performed histopathology analysis. With all infection data, we revised that CnHsf3 plays a moderate but consistent role in fungal pathogenicity, particularly during pulmonary infection. The moderate but consistent virulence attenuation might be more due to the fact that multiple parallel potent ROS detoxification machineries are at work, including mtSODs, GSH, reductases that remain functional to compensate for the loss of CnHsf3 function than the temperature response. We have provided these new findings and elaborated their implications in the revised text. Please see lines 123-150 and 599-608.

BALB/c mice are INBRED models (<https://www.jax.org/strain/000651>) commonly used in pathogenicity studies of *Cryptococcus neoformans*¹⁻⁴, similar to AJC/r, and the results often agree upon each other. Indeed, the data from our BALB/c model study is consistent an earlier study using AJC/r mice⁵.

The HSF family proteins are known as the master regulators of UPR in all model organisms examined and the regulation of HSF in mitochondrial is largely unknown. Our study, utilized a pathogenic fungal model, allowed us to discover a noncanonical function of HSF proteins that will have important implications to studies of other model organisms.

1. Hole, C.R. *et al.* Induction of memory-like dendritic cell responses in vivo. *Nat Commun* **10**, 2955 (2019).
2. Bloom, A.L.M. *et al.* Thermotolerance in the pathogen *Cryptococcus neoformans* is linked to antigen masking via mRNA decay-dependent reprogramming. *Nat Commun* **10**, 4950 (2019).
3. Li, Z. *et al.* Cisplatin protects mice from challenge of *Cryptococcus neoformans* by targeting the Prp8 intein. *Emerg Microbes Infect* **8**, 895-908 (2019).
4. Chow, E.W. *et al.* Elucidation of the calcineurin-Crz1 stress response transcriptional network in the human fungal pathogen *Cryptococcus neoformans*. *PLoS Genet* **13**, e1006667 (2017).
5. Jung, K.W. *et al.* Systematic functional profiling of transcription factor networks in *Cryptococcus neoformans*. *Nat Commun* **6**, 6757 (2015).

Reviewer #3 (Remarks to the Author):

None.

Response: We are grateful for the very positive response.

Reviewer #4 (Remarks to the Author):

The authors were extremely responsive to previous reviewer's comments and the manuscript has significantly improved. The authors included several new experiments, including pathogenicity studies in mice, repeated a large number of critical experiments to validate their conclusions and increased the rigor of the experiments by including proper controls and providing appropriate statistical analyses.

This reviewer is highly satisfied with the new revised version of the manuscript and

congratulate the authors for the enormous effort put towards characterizing CnHsf3.

Response: We are grateful for the positive comments. In the revised manuscript, we have modified the text to further improve our manuscript. Please find our responses below.

Only two minor comments:

Figure S2 A and C are extremely small and they can't be read.

Reponses: We thank for the comment. We have enlarged the figure and labels. Please see Figure S2.

As part of the response to the reviewers, the authors indicated that antimycin A treated Cnhsf3 mutant cells had about 50% of induction in mtROS and they hypothesized that antimycin A does not fully recapitulate the response from heat stress.” However, text in the revised manuscript lines 302-304 state that “antimycin A treatment fully recapitulated the phenotypes of cells under high temperatures”. Please clarify these contradicting statements.

Reponses: We thank the reviewer for the comment. The antimycin A-treated *Cnhsf3Δ* mimics the regulation patterns from the heat shock response, including induction in ROS production, gene expression and cell growth defects. However, high temperature is a more potent mtROS inducer in *Cnhsf3Δ* cells than antimycin A treatment. Therefore, we fully agree with this comment that antimycin A cannot FULLY recapitulate the heat response, but only mimics the regulation patterns and phenotypes. We have clarified this contradicting statement, to more clearly and precisely describe our finding in the revised manuscript. Please see lines 310-312.

REVIEWERS' COMMENTS

Reviewer #2 (Remarks to the Author):

In the second revised manuscript, the authors fully responded to my comments by performing additional animal study with two independent mutants and a complemented strain. It is also nice for the authors to include fungal burden assay and histopathological examination data. These new data support that Hsf3 plays a very minor but consistent role in the virulence of *Cryptococcus neoformans*. I do not have more comments.

Reviewer #2 (Remarks to the Author):

In the second revised manuscript, the authors fully responded to my comments by performing additional animal study with two independent mutants and a complemented strain. It is also nice for the authors to include fungal burden assay and histopathological examination data. These new data support that Hsf3 plays a very minor but consistent role in the virulence of *Cryptococcus neoformans*. I do not have more comments.

Response: We sincerely appreciate the positive comment from the reviewer.